# ANTN: Bridging Autoregressive Neural Networks and Tensor Networks for Quantum Many-Body Simulation

**Zhuo Chen**[123]    **Laker Newhouse**[4*]   **Eddie Chen**[3*]   **Di Luo**[1235†]   **Marin Soljačić**[13]

[1]NSF AI Institute for Artificial Intelligence and Fundamental Interactions
[2]Center for Theoretical Physics, Massachusetts Institue of Technology
[3]Department of Physics, Massachusetts Institute of Technology
[4]Department of Mathematics, Massachusetts Institute of Technology
[5]Department of Physics, Harvard University
`{chenzhuo,lakern,ezchen,diluo,soljacic}@mit.edu`

## Abstract

Quantum many-body physics simulation has important impacts on understanding fundamental science and has applications to quantum materials design and quantum technology. However, due to the exponentially growing size of the Hilbert space with respect to the particle number, a direct simulation is intractable. While representing quantum states with tensor networks and neural networks are the two state-of-the-art methods for approximate simulations, each has its own limitations in terms of expressivity and inductive bias. To address these challenges, we develop a novel architecture, Autoregressive Neural TensorNet (ANTN), which bridges tensor networks and autoregressive neural networks. We show that Autoregressive Neural TensorNet parameterizes normalized wavefunctions, allows for exact sampling, generalizes the expressivity of tensor networks and autoregressive neural networks, and inherits a variety of symmetries from autoregressive neural networks. We demonstrate our approach on quantum state learning as well as finding the ground state of the challenging 2D $J_1$-$J_2$ Heisenberg model with different systems sizes and coupling parameters, outperforming both tensor networks and autoregressive neural networks. Our work opens up new opportunities for quantum many-body physics simulation, quantum technology design, and generative modeling in artificial intelligence.

## 1   Introduction

Quantum many-body physics is fundamental to our understanding of the universe. It appears in high energy physics where all the fundamental interactions in the Standard Model, such as quantum electrodynamics (QED) and quantum chromodynamics (QCD), are described by quantum mechanics. In condensed matter physics, quantum many-body physics has led to a number of rich phenomena and exotic quantum matters, including superfluids, superconductivity, the quantum Hall effect, and topological ordered states (Girvin & Yang, 2019). As an application, quantum many-body physics is crucial for new materials design. The electronic structure problem and chemical reactions in quantum chemistry are governed by quantum many-body physics. The recent development of quantum computers is also deeply connected to quantum many-body physics. A multi-qubit quantum device is intrinsically a quantum many-body system, such that progress on quantum computer engineering is tied to our understanding of quantum many-body physics (Preskill, 2021).

---

[*]equal contribution

[†]corresponding author: diluo@mit.edu

37th Conference on Neural Information Processing Systems (NeurIPS 2023).

All information of a closed quantum many-body system is captured by the wavefunction, whose properties are described by the famous Schrödinger equation. An important tool to study and understand quantum many-body physics is to classically simulate the wavefunction. However, the wavefunction is a high dimensional function in Hilbert space, whose dimension grows exponentially with the number of particles. For example, for qubit systems where each qubit has two degrees of freedom), the wavefunction of 300 qubits will have dimension $2^{300}$, which is larger than the number of atoms in the observable universe. Furthermore, the Schrödinger equation is a complex-valued high-dimensional equation, which is challenging to solve or simulate in general.

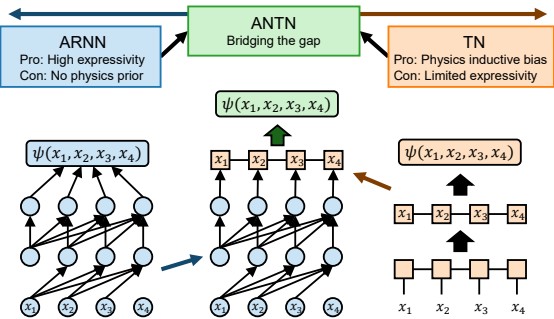

Figure 1: Diagrammatic representation of autoregressive neural network (ARNN), tensor network (TN) and our Autoregressive Neural TensorNet (ANTN).

A number of algorithms have been developed to simulate quantum many-body physics, including quantum Monte Carlo, tensor networks, neural network quantum states, and quantum computation. In particular, computing the ground state of quantum many-body systems is of great interest. One important approach is the variational principle, which provides an upper bound for the ground state energy. To apply the variational principle successfully, one must design an ansatz that can represent and optimize the wavefunction efficiently. Tensor networks and neural network quantum states (Carleo & Troyer, 2017) are the two main state-of-the-art methods that can be applied with the variational principle for quantum many-body simulation. However, tensor networks usually suffer from an expressivity issue in systems with more than one dimension, while neural network quantum states usually lack inductive bias from the underlying physics structure and have sign structure challenges in the representation.

In this paper, we develop a novel architecture, Autoregressive Neural TensorNet (ANTN), to bridge neural network quantum states and tensor networks, achieving the best of both worlds. In particular, our contributions are threefold:

- Develop ANTN with two variants called "elementwise" and "blockwise," which each naturally generalize the two state-of-the-arts ansatzes, tensor networks (TN) and autoregressive neural networks (ARNN), to provide proper inductive bias and high expressivity.
- Prove that ANTN is normalized with exact sampling, has generalized expressivity of TN and ARNN, and inherits multiple symmetries from ARNN.
- Demonstrate our methods on quantum state learning and variationally finding the ground state of the challenging 2D $J_1$-$J_2$ Heisenberg model, outperforming both TN and ARNN.

## 2 Related Work

**Tensor Networks (TN)** represent high-dimensional wavefunctions using low-rank tensor decomposition, notably matrix product state (MPS) (Vidal, 2003, 2004), PEPS (Verstraete & Cirac, 2004), and MERA (Vidal, 2007). They capture the entanglement structure of physical systems and, with algorithms like the density matrix renormalization group (DMRG) (White, 1992), are used for state simulations and real-time dynamics. However, their expressivity can be limited in systems with more than one dimension and systems with high entanglement. In machine learning, TN appears as tensor train (Oseledets, 2011) and CP decomposition methods.

**Neural Network Quantum State (NNQS)** leverages neural networks for high dimensional wavefunction representation (Carleo & Troyer, 2017). It's been demonstrated that many quantum states can be approximated or represented by NNQS (Sharir et al., 2021; Gao & Duan, 2017; Lu et al., 2019; Levine et al., 2019a; Luo et al., 2021b,a; Deng et al., 2017; Huang & Moore, 2021; Vieijra et al., 2020), and it has yielded state-of-the-art results in computing quantum system properties (Gutiérrez & Mendl, 2020; Schmitt & Heyl, 2020; Vicentini et al., 2019; Yoshioka & Hamazaki, 2019; Hartmann & Carleo, 2019; Nagy & Savona, 2019; Luo et al., 2021b, 2022a). Key advancements in NNQS include ARNN that improves sample efficiency and gradient estimation, and the development of

neural networks that adhere to the underlying symmetries of quantum systems (Luo et al., 2021b; Hibat-Allah et al., 2020; Choo et al., 2019; Luo & Clark, 2019; Hermann et al., 2019; Pfau et al., 2020; Luo et al., 2021a, 2022b, 2021b; Chen et al., 2022). Despite its potential, NNQS faces challenges such as the lack of physics prior and the sign structure problem. While there are attempts to integrate NNQS with TN, including matrix product state with neural network backflow (Lami et al., 2022) and generalizing MPS to RNN (Wu et al., 2022), the former can not produce normalized wavefunctions, while the latter only builds on tensor contractions that lacks the nonlinear activation functions of standard neural network wavefunctions.

## 3 Background

### 3.1 Quantum Preliminaries

In this work, we focus on qubit systems in quantum many-body physics. The wavefunction or quantum state $\psi$ of the system is a normalized function $\psi : \mathbb{Z}_2^n \to \mathbb{C}$ with $\sum_{\boldsymbol{x}} |\psi(\boldsymbol{x})|^2 = 1$, where $n$ is the system size. The input to the wavefunction is an $n$-bit string $\boldsymbol{x} \in \{0, 1\}^n$. Therefore, the wavefunction $\psi$ can be viewed as a complex-valued vector of size $2^n$, with Dirac notation $\langle\psi|$ and $|\psi\rangle$ correspond to a conjugate row vector and a column vector respectively, and $\psi(\boldsymbol{x}) = \langle\boldsymbol{x}|\psi\rangle$. Because the size of the wavefunction grows exponentially with the system size $n$, a direct computation quickly becomes intractable as the system size increases. The goal of NNQS is to design a compact architecture that can approximate and optimize the wavefunction efficiently.

### 3.2 Quantum State Learning

Given a quantum state $|\phi\rangle$, we are often interested in finding $|\psi_\theta\rangle$ that is closest to $|\phi\rangle$ given the variational ansatz. In quantum mechanics, the closeness of two quantum states is measured by the quantum fidelity $F = |\langle\phi|\psi_\theta\rangle|^2 = |\sum_{\boldsymbol{x}} \phi^*(\boldsymbol{x})\psi_\theta(\boldsymbol{x})|^2$ where $^*$ refers to complex conjugation. Quantum fidelity satisfies $0 \leq F \leq 1$, with $F = 1$ corresponding to exact match, and $F = 0$ orthogonal quantum states. Finding $F_{\max}$ for a given ansatz allows us to quantify how good the ansatz can be used to approximate that state. In practice, minimizing $-\log F$ is usually better than maximizing $F$ itself. This can be achieved by enumerating all the basis $\boldsymbol{x}$ in small systems and stochastically in large systems such as quantum state tomography (see Appendix A.1).

### 3.3 Variational Monte Carlo

For a given quantum system with $n$ qubits, the Hamiltonian $\hat{\mathcal{H}}$ can be written as a Hermitian matrix of size $2^n \times 2^n$. The ground state energy $E_g$ of the system is the smallest eigenvalue of $\hat{\mathcal{H}}$ and the ground state is the corresponding eigenvector. For large system sizes, finding the ground state directly is usually impossible. In this case, the variational principle in quantum mechanics provides an upper bound on the ground state energy $E_g$. For all (normalized) wavefunctions $|\psi\rangle$, it is evidental that $E_g \leq \langle\psi|\hat{\mathcal{H}}|\psi\rangle$. Therefore, finding the $|\psi_\theta\rangle$ that minimizes $E_\theta = \langle\psi_\theta|\hat{\mathcal{H}}|\psi_\theta\rangle$ gives the lowest upper bound of the ground state energy. For large system sizes, we can stochastically evaluate and minimize

$$\langle\psi_\theta|\hat{\mathcal{H}}|\psi_\theta\rangle = \sum_{\boldsymbol{x}\boldsymbol{x}'} |\psi_\theta(\boldsymbol{x})|^2 \frac{\mathcal{H}_{\boldsymbol{x},\boldsymbol{x}'}\psi_\theta(\boldsymbol{x}')}{\psi_\theta(\boldsymbol{x})} = \mathbb{E}_{\boldsymbol{x}\sim|\psi_\theta|^2} \frac{\sum_{\boldsymbol{x}'} \mathcal{H}_{\boldsymbol{x},\boldsymbol{x}'}\psi_\theta(\boldsymbol{x}')}{\psi_\theta(\boldsymbol{x})}, \qquad (1)$$

where $\mathcal{H}_{\boldsymbol{x},\boldsymbol{x}'}$ refers to the matrix element of $\hat{\mathcal{H}}$ and we interpret $|\psi_\theta(\boldsymbol{x})|^2$ as a probability distribution. The summation over $\boldsymbol{x}'$ can be efficiently computed given $\boldsymbol{x}$ since the Hamiltonian is usually sparse. The gradient $\nabla_\theta E_\theta$ can also be calculated stochastically in a similar fashion (see Appendix A.2).

## 4 Method

### 4.1 Preliminaries

**Autoregressive Neural Network Wavefunction.** The autoregressive neural network (ARNN) (Fig. 1 left) parameterizes the full probability distribution as a product of conditional probability distributions

as

$$p(\boldsymbol{x}) = \prod_{i=1}^{n} p(x_i|\boldsymbol{x}_{<i}), \tag{2}$$

where $\boldsymbol{x} = (x_1, \dots, x_n)$ is a configuration of $n$ qubits and $\boldsymbol{x}_{<i} = (x_1, \dots, x_{i-1})$ is any configuration before $x_i$. The normalization of the full probability can be guaranteed from the normalization of individual conditional probabilities. The ARNN also allows for exact sampling from the full distribution by sampling sequentially from the conditional probabilities (see Appendix B.1). The autoregressive constructions for probabilities can be easily modified to represent quantum wavefunction by (1) replacing $p(x_i|\boldsymbol{x}_{<i})$ with a complex-valued conditional wavefunction $\psi(x_i|\boldsymbol{x}_{<i})$ and (2) using the following normalization condition for the conditional wavefunctions: $\sum_x |\psi(x_i|\boldsymbol{x}_{<i})|^2 = 1$ (Sharir et al., 2020; Hibat-Allah et al., 2020; Luo et al., 2021b). Similar to the case of probabilities, ARNN automatically preserves the normalization of wavefunctions and allows for exact sampling (see Appendix B.1). Because of this, ARNN is often more efficient when training and computing various quantum observables compared to other generative neural networks.

**Matrix Product State (MPS).** MPS (also known as tensor train) is a widely used TN architecture to study quantum many-body physics (Vidal, 2003, 2004). The MPS defines a wavefunction using $n$ rank-3 tensors $M_{x_i}^{\alpha_{i-1}\alpha_i}$ with $\alpha_{i-1}$ (or $\alpha_i$) the index of the left (or right) bond dimensions and $x_i$ the configuration of the $i$th qubit. Then, the full wavefunction is generated by first choosing a particular configuration $\boldsymbol{x} = (x_1, \dots, x_n)$ and then contracting the tensors selected by this configuration (Fig. 1 right) as

$$\psi(\boldsymbol{x}) = \sum_{\alpha_1,\dots,\alpha_{n-1}} M_{x_1}^{\alpha_0\alpha_1} \cdots M_{x_n}^{\alpha_{n-1}\alpha_n} = \sum_{\boldsymbol{\alpha}} \prod_{i=1}^{n} M_{x_i}^{\alpha_{i-1}\alpha_i}, \tag{3}$$

where the left-and-right-most bond dimensions are assumed to be $\mathcal{D}(\alpha_0) = \mathcal{D}(\alpha_{n+1}) = 1$ so are not summed.

## 4.2 Autoregressive Neural TensorNet (ANTN)

**Elementwise and Blockwise Construction for ANTN.** Both ARNN and MPS are powerful ansatzes for parameterizing quantum many-body wavefunctions. Albeit very expressive, ARNN lacks the physics prior of the system of interest. In addition, since wavefunctions are complex-valued in general, learning the sign structure can be a nontrivial task (Westerhout et al., 2020). MPS, on the other hand, contains the necessary physics prior and can efficiently represent local or quasi-local sign structures, but its expressivity is severely limited. The internal bond dimension needs to grow exponentially to account for a linear increase in entanglement. It turns out that MPS representation is not unique in that many MPS actually represent the same wavefunction; and if we choose a particular constraint (without affecting the expressivity), MPS allows for efficient evaluation of conditional probability and exact sampling (Ferris & Vidal, 2012) (see Appendix B.2). Because both these ansatzes allow for exact sampling, it is natural to combine them to produce a more powerful ansatz. Therefore, we develop the Autoregressive Neural TensorNet (ANTN) (Fig. 1 (middle)). In the last layer of the ANTN, instead of outputting the conditional wavefunction $\psi(x_i|\boldsymbol{x}_{<i})$, we output a conditional wavefunction tensor $\tilde{\psi}^{\alpha_{i-1}\alpha_i}(x_i|\boldsymbol{x}_{<i})$ for each site. Defining the left partially contracted tensor up to qubit $j$ as $\tilde{\psi}_L^{\alpha_j}(\boldsymbol{x}_{\leq j}) := \sum_{\boldsymbol{\alpha}_{<j}} \prod_{i=1}^{j} \tilde{\psi}^{\alpha_{i-1}\alpha_i}(x_i|\boldsymbol{x}_{<i})$, we can define the (unnormalized) marginal probability distribution as

$$q(\boldsymbol{x}_{\leq j}) := \sum_{\alpha_j} \tilde{\psi}_L^{\alpha_j}(\boldsymbol{x}_{\leq j}) \tilde{\psi}_L^{\alpha_j}(\boldsymbol{x}_{\leq j})^*, \tag{4}$$

where $*$ denotes complex conjugation. Then, the (normalized) conditional probability can be obtained as $q(x_j|\boldsymbol{x}_{<j}) = q(\boldsymbol{x}_{\leq j})/\sum_{x_j} q(\boldsymbol{x}_{\leq j})$. We construct the overall wavefunction by defining both its amplitude and phase according to

$$\psi(\boldsymbol{x}) := \sqrt{q(\boldsymbol{x})} e^{i\phi(\boldsymbol{x})}, \tag{5}$$

with $q(\boldsymbol{x}) =: \prod_{i=1}^{n} q(x_i|\boldsymbol{x}_{<i})$ and the phase $\phi(\boldsymbol{x}) =: \mathrm{Arg} \sum_{\boldsymbol{\alpha}} \prod_{i=1}^{n} \tilde{\psi}^{\alpha_{i-1},\alpha_i}(x_i|\boldsymbol{x}_{<i})$. In other words, we define the amplitude of the wavefunction through the conditional probability distributions and define the phase analogous to the standard MPS.

We develop two different constructions for ANTN that differ in the last layer on how to construct conditional wavefunction tensors.

The elementwise ANTN is given by

$$\tilde{\psi}^{\alpha_{i-1}\alpha_i}(x_i|\boldsymbol{x}_{<i}) = M_{x_i}^{\alpha_{i-1}\alpha_i} + f_{NN}(x_i, \alpha_{i-1}, \alpha_i|\boldsymbol{x}_{<i}), \tag{6}$$

where $f_{NN}(x_i, \alpha_{i-1}, \alpha_i|\boldsymbol{x}_{<i})$ is the complex-valued output. The blockwise ANTN is given by

$$\tilde{\psi}^{\alpha_{i-1}\alpha_i}(x_i|\boldsymbol{x}_{<i}) = M_{x_i}^{\alpha_{i-1}\alpha_i} + f_{NN}(x_i|\boldsymbol{x}_{<i}), \tag{7}$$

where the complex-valued output $f_{NN}(x_i|\boldsymbol{x}_{<i})$ is broadcasted over $\alpha$. This results in a lower complexity that allows us to use a larger maximum bond dimension.

**Transformer and PixelCNN Based ANTN.** Our construction above is general and can be applied to any standard ARNN. Depending on the application, we can use different ARNN architectures. In this work, we choose the transformer and PixelCNN depending on the specific tasks. The transformer (Vaswani et al., 2017) used here is similar to a decoder-only transformer implemented in (Luo et al., 2020), and The PixelCNN we use is the gated PixelCNN (Van den Oord et al., 2016) implemented in (Chen et al., 2022).

**MPS Initialization.** Since our ANTN generalizes from TN, both the elementwise and the blockwise can take advantage of the optimized MPS from algorithms such as DMRG (White, 1992) as an initialization. In practice, we can initialize the TN component with the optimized DMRG results of the same bond dimension (similar to (Lami et al., 2022; Wu et al., 2022)). The MPS Initialization can also be thought of as a pretraining of the ANTN. This is a nice feature since it provides a good sign structure and inductive bias from the physics structure, which does not exist in the conventional ARNN.

**Limitations.** In this work, we only integrated MPS into the ANTN, where MPS may not be the best TN in various settings. Besides MPS, many other TNs also allow efficient evaluation of conditional probabilities and exact sampling, such as MERA (Vidal, 2007) and PEPS (Verstraete & Cirac, 2004), or cylindrical MPS for periodic boundary conditions. In fact, the recently developed TensorRNN (Wu et al., 2022) can also be viewed as a type of TN and can be integrated into our construction for future work. In addition, while the ANTN construction is general in terms of the base ARNN used, our current study only focuses on transformer and PixelCNN. Lastly, as shown later in Sec. 5.1, the current implementation of ANTN has an additional sampling overhead that is linear in system size which can be avoided.

## 5 Theoretical Results

### 5.1 Exact Sampling and Complexity Analysis of ANTN

**Theorem 5.1.** *Autoregressive Neural TensorNet wavefunction is automatically normalized and allows for exact sampling.*

*Proof.* This is a direct consequence that we defined the amplitude of the wavefunction through normalized conditional probability distributions $q(x_i|\boldsymbol{x}_{<i})$. (See Appendix B.1 for the detailed sampling procedure.) $\square$

**Complexity Analysis.** We first note that for MPS, the number of parameters and computational complexity for evaluating a bitstring $\boldsymbol{x}$ scales as $\mathcal{O}(n\chi^2)$, where $n$ is the number of particles and $\chi = \mathcal{D}(\alpha)$ is the (maximum) bond dimension of MPS. The sampling complexity scales as $\mathcal{O}(n^2\chi^2)$. The DMRG algorithm has a computational complexity of $\mathcal{O}(n\chi^3)$ (White, 1992). The number of parameters and computational complexity of ARNN depends on the specific choice. Assuming the ARNN has $n_{\mathrm{ARNN}}$ parameters and a single pass (evaluation) of the ARNN has a computational complexity of $c_{\mathrm{ARNN}}$, then the sampling complexity of the ARNN is $\mathcal{O}(nc_{\mathrm{ARNN}})$ in our current implementation. The ANTN based on such ARNN would have $\mathcal{O}(n_{\mathrm{ARNN}} + n\chi^2 h_{\mathrm{dim}})$ parameters for the elementwise construction, and $\mathcal{O}(n_{\mathrm{ARNN}} + n\chi^2 + nh_{\mathrm{dim}})$ parameters for the blockwise construction. The computational complexity for evaluating and sampling bitstrings scales as $\mathcal{O}(n^s(c_{\mathrm{ARNN}} + n\chi^2 h_{\mathrm{dim}}))$ for elementwise construction and $\mathcal{O}(n^s(c_{\mathrm{ARNN}} + n\chi^2))$ for blockwise construction with $s = 0$ for evaluation and $s = 1$ for sampling.

We note that the complexity analysis above is based on our current implementation, where the sampling procedure for all algorithms has an $\mathcal{O}(n)$ overhead compared to evaluation. It is possible to remove it by storing the partial results during the sampling procedure.

Then, each gradient update of the variational Monte Carlo algorithm requires sampling once and evaluating $N_{\mathrm{con}}$ times where $N_{\mathrm{con}}$ is the number of connected (non-zero) $\boldsymbol{x}'$'s in $\mathcal{H}_{\boldsymbol{x},\boldsymbol{x}'}$ given $\boldsymbol{x}$, which usually scales linearly with the system size. Then, the overall complexity is $\mathcal{O}(N_s(N_{\mathrm{con}}c_{\mathrm{eva}}+c_{\mathrm{samp}}))$ with $N_s$ the batch size, $c_{\mathrm{eva}}$ the evaluation cost and $c_{\mathrm{samp}}$ the sampling cost. Usually, the first part dominates the cost.

The difference in computational complexities and number of parameters between elementwise ANTN and blockwise ANTN implies that blockwise ANTN is usually more economical than elementwise ANTN for the same hidden dimsnieon $h_{\mathrm{dim}}$ and bond dimension $\chi$. Therefore, for small bond dimensions, we use the elementwise ANTN for a more flexible parameterization with a higher cost. In contrast, for large bond dimensions, we use the blockwise ANTN, which saves computational complexity and improves the initial performance (with DMRG initialization) of the blockwise ANTN at the cost of less flexible modifications from the ARNN. We also note that compared to the state-of-the-art MPS simulation, even for our blockwise ANTN, a small bond dimension usually suffices. For example, in the later experimental section, we use bond dimension 70 for blockwise ANTN while the best MPS results use bond dimension 1024, which has much more parameters than our ANTN. In general, our ANTN has much fewer parameters compared to MPS due to the $\mathcal{O}(\chi^2)$ parameters scaling which dominates at large bond dimension.

## 5.2 Expressivity Results of ANTN

**Theorem 5.2.** *Autoregressive Neural TensorNet can have volume law entanglement, which is strictly beyond the expressivity of matrix product states.*

*Proof.* We proved in Thm. 5.3 that ANTN can be reduced to an ARNN, which has been shown to have volume law entanglement that cannot be efficiently represented by TN (Sharir et al., 2021; Levine et al., 2019b). Hence, ANTN can represent states that cannot in general be represented by MPS efficiently; thus it has strictly greater expressivity. □

**Theorem 5.3.** *Autoregressive Neural TensorNet has generalized expressivity over tensor networks and autoregressive neural networks.*

*Proof.* Thm B.2of Appendix B.4 shows that both TN and ARNN are special cases as ANTN and Thm B.3of Appendix B.4 shows that ANTN can be written as either a TN or an ARNN with an exponential (in system size) number of parameters. Thus, ANTN generalizes the expressivity over both TN and ARNN. □

## 5.3 Symmetry Results of ANTN

Symmetry plays an important role in quantum many-body physics and quantum chemistry. Many of the symmetries can be enforced in ARNN via the two classes of the symmetries—*mask symmetry* and *function symmetry*.

**Definition 5.1** (Mask Symmetry)**.** A conditional wavefunction $\psi(x_i|\boldsymbol{x}_{<i})$ has a *mask symmetry* if $\psi(x_i|\boldsymbol{x}_{<i}) = 0$ for some $x_i$ given $\boldsymbol{x}_{<i}$.

**Definition 5.2** (Function Symmetry)**.** A conditional wavefunction tensor $\psi(x_i|\boldsymbol{x}_{<i})$ has a *function symmetry* over a function $F$ if $\psi(x_i|\boldsymbol{x}_{<i}) = \psi(F(x_i;\boldsymbol{x}_{<i})|\boldsymbol{F}(\boldsymbol{x}_{<i}))$ where $\boldsymbol{F}(\boldsymbol{x}_{\leq i}) := \{F(x_1), F(x_2; x_1), \ldots, F(x_i; x_{<i})\}$.

Here, we list several symmetries that ANTN inherits from ARNN and show the proofs in Appendix B.4.

**Theorem 5.4.** *Autoregressive Neural TensorNet inherits mask symmetry and function symmetry from autoregressive neural networks.*

**Corollary 5.4.1** (Global U(1) Symmetry)**.** *Autoregressive Neural TensorNet can realize global U(1) symmetry, which conserves particle number.*

**Corollary 5.4.2** ($\mathbb{Z}_2$ Spin Flip Symmetry)**.** *Autoregressive Neural TensorNet can realize $\mathbb{Z}_2$ spin flip symmetry such that the wavefunction is invariant under conjugation of the input.*

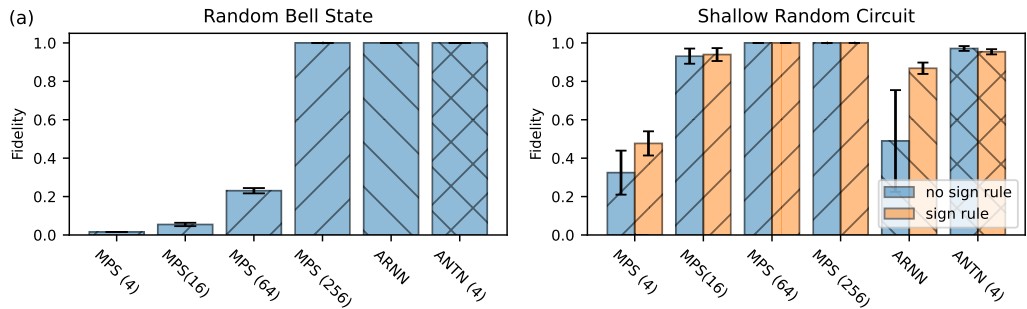

Figure 2: Fidelity ↑ on quantum state learning with 16 qubits for TN (MPS), ARNN (transformer) and ANTN (elementwise construction with transformer+MPS). (a) Learning random Bell states. (b) Learning real-valued depth-4 random circuit with and without sign rule. The error bar denotes the standard deviation (not the standard error of the mean) of the fidelities over the random states sampled from the corresponding distribution. The mean and standard deviation are calculated from 10 random states. The numbers inside the parentheses denote the bond dimension.

**Corollary 5.4.3** (Discrete Abelian and Non-Abelian Symmetries). *Autoregressive Neural TensorNet can realize discrete Abelian and Non-Abelian symmetries.*

# 6    Experiments

## 6.1    Quantum State Learning

As stated previously, ANTN generalizes both ARNN and TN to take advantage of both the expressivity and the inductive bias. Here, we test the ANTN on learning physically important quantum states to demonstrate this ability. In this task, we use the transformer neural network for ARNN and ANTN due to the 1D structure of the system.

**Experiments on expressivity.** We first test the expressivity of the ANTN by learning a class of well-known high-entangled states—random permuted Bell states. A 2-qubit Bell state is defined as $(|00\rangle + |11\rangle)/\sqrt{2}$, which has 1 bit of entanglement. For a system size of $n$, we randomly pair qubits in the first half of the system with the qubits in the second half of the system to be Bell states. This process creates quantum states with $n/2$ bit of entanglement between the first and second half of the system. It can be shown that a bond dimension of $2^{n/2}$ is required for MPS to fully represent such a system. In Fig. 2 (a), we plot the quantum fidelity of learning 16-qubit random Bell states. As the figure shows, MPS cannot represent the states without reaching the required bond dimension of $256 = 2^8$. The ARNN and ANTN, on the other hand, can represent such states without limitations from the bond dimension.

**Experiments on inductive bias.** We then test the physics inductive bias of the ANTN. One of the physics inductive biases of MPS is that it can represent wavefunctions with local or quasi-local sign structures that can be hard for neural networks to learn. These wavefunctions can have a fluctuating sign depending on the configuration $x$. Here, we use (real-valued) shallow random circuits to mimic the short-range interaction and generate states with sign structures. We test the algorithms on these states both with and without the "sign rule", which means that we explicitly provide the sign structure to the algorithm. As shown in Fig. 2 (b), the fidelity of MPS only depends weakly on the sign rule, whereas the fidelity of ARNN can change drastically. Our ANTN inherits the property of MPS and is thus not affected by the sign structure. Furthermore, being more expressive, ANTN with a bond dimension of 4 already performs better than MPS with a bond dimension of 16.

## 6.2    Variational Monte Carlo

We further test our algorithm on finding the ground state of the challenging 2D $J_1$-$J_2$ Heisenberg model with open boundary condition. The model has a rich phase diagram with at least three different phases across different $J_2/J_1$ values (Capriotti et al., 2004) (Lante & Parola, 2006). In addition, the

| Energy per site ↓ $8 \times 8$ | | | |
|---|---|---|---|
| Algorithms | $J_2 = 0.2$ | $J_2 = 0.5$ | $J_2 = 0.8$ |
| RBM (NS) | -1.9609(16) | -1.5128(20) | -1.7508(19) |
| RBM (S) | -2.1944(17) | -1.8625(20) | -0.7267(29) |
| PixelCNN (NS) | -2.21218(16) | -1.77058(29) | -1.93825(16) |
| PixelCNN (S) | -2.23171(9) | -1.88902(19) | -1.83672(26) |
| Elementwise (8 NS) | **-2.23690(4)** | *-1.93018(8)* | *-2.00036(16)* |
| Elementwise (8 S) | *-2.23688(4)* | **-1.93102(7)** | **-2.00262(11)** |
| Blockwise (70 NS) | -2.23484(7) | -1.93000(7) | -1.99148(13) |
| Blockwise (70 S) | -2.23517(6) | -1.92880(9) | -1.99246(13) |

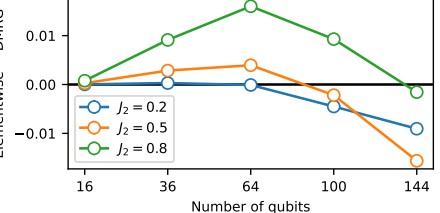

Table 1: Energy per site ↓ for $8 \times 8$ system with various algorithms where elementwise and blockwise are two constructions of ANTN (with PixelCNN + MPS). The bond dimensions for ANTN are labeled inside the parentheses. For each algorithm, we test it both with the sign rule (S) and without the sign rule (NS) The best energy is highlighted in boldface and the second in italic.

Figure 3: Energy per site difference ↓ between elementwise ANTN with bond dimension 8 and MPS with bond dimension 1024 optimized using DMRG algorithm for various $J_2$ and different system sizes from $4 \times 4$ to $12 \times 12$.

| Energy per site ↓ $10 \times 10$ | | | | | | | |
|---|---|---|---|---|---|---|---|
| Algorithms | $J_2 = 0.2$ | $J_2 = 0.3$ | $J_2 = 0.4$ | $J_2 = 0.5$ | $J_2 = 0.6$ | $J_2 = 0.7$ | $J_2 = 0.8$ |
| MPS (8) | -1.997537 | -1.893753 | -1.797675 | -1.734326 | -1.716253 | -1.768225 | -1.869871 |
| MPS (70) | -2.191048 | -2.069029 | -1.956480 | -1.866159 | -1.816249 | -1.854296 | -2.007845 |
| MPS (1024) | -2.255633 | -2.138591 | -2.031681 | *-1.938770* | **-1.865561** | **-1.894371** | **-2.062730** |
| PixelCNN | -2.22462(24) | -2.12873(14) | -2.02053(14) | -1.74098(29) | -1.71885(27) | -1.81800(13) | -1.98331(17) |
| Elementwise (8) | **-2.26034(6)** | **-2.14450(4)** | **-2.03727(7)** | **-1.94001(6)** | *-1.85684(10)* | *-1.88643(7)* | *-2.05707(43)* |
| Blockwise (70) | *-2.25755(8)* | *-2.14152(8)* | *-2.03319(10)* | -1.93842(42) | -1.85270(12) | -1.87853(13) | -2.05088(14) |
| Energy per site ↓ $12 \times 12$ | | | | | | | |
| Algorithms | $J_2 = 0.2$ | $J_2 = 0.3$ | $J_2 = 0.4$ | $J_2 = 0.5$ | $J_2 = 0.6$ | $J_2 = 0.7$ | $J_2 = 0.8$ |
| MPS (8) | -1.998207 | -1.887531 | -1.800784 | -1.735906 | -1.720619 | -1.788652 | -1.893916 |
| MPS (70) | -2.185071 | -2.059443 | -1.944832 | -1.851954 | -1.812450 | -1.853650 | -2.030131 |
| MPS (1024) | *-2.264084* | -2.141043 | -2.027736 | -1.931243 | **-1.858846** | *-1.913483* | *-2.093013* |
| PixelCNN | -2.24311(102) | -2.12616(23) | -2.01768(21) | -1.74282(30) | -1.72637(16) | -1.85239(13) | -2.03226(59) |
| Elementwise (8) | **-2.27446(27)** | **-2.15537(6)** | **-2.04437(7)** | **-1.94686(6)** | *-1.85443(15)* | **-1.91391(10)** | **-2.09457(10)** |
| Blockwise (70) | -2.26152(50) | *-2.15395(7)* | *-2.04225(8)* | *-1.94298(43)* | -1.85176(15) | -1.90571(12) | -2.09113(43) |

Table 2: Energy per site ↓ for $10 \times 10$ and $12 \times 12$ system with various algorithms where elementwise and blockwise are two constructions of ANTN (with PixelCNN + MPS). The bond dimensions for MPS and ANTN are labeled in parentheses. The MPS is optimized with DMRG algorithm. The best energy is highlighted in boldface and the second best value is highlighted in italic.

complicated structure of its ground state makes it a robust model on which to test state-of-the-art methods. Here, we use the PixelCNN for ARNN and ANTN due to the 2D geometry of the system.

The 2D $J_1$-$J_2$ Hamiltonian is given by

$$\hat{\mathcal{H}} = J_1 \sum_{\langle i,j \rangle} \vec{\sigma_i} \cdot \vec{\sigma_j} + J_2 \sum_{\langle\langle i,j \rangle\rangle} \vec{\sigma_i} \cdot \vec{\sigma_j}, \tag{8}$$

where subscript refers to the site of the qubits, $\langle \cdot, \cdot \rangle$ is the nearest neighbour and $\langle\langle \cdot, \cdot \rangle\rangle$ is the next nearest neighbour. $\vec{\sigma_i} \cdot \vec{\sigma_j} = X_i \otimes X_j + Y_i \otimes Y_j + Z_i \otimes Z_j$ with $X$, $Y$ and $Z$ the Pauli matrices

$$X = \begin{bmatrix} 0 & 1 \\ 1 & 0 \end{bmatrix}, \quad Y = \begin{bmatrix} 0 & -i \\ i & 0 \end{bmatrix}, \quad Z = \begin{bmatrix} 1 & 0 \\ 0 & -1 \end{bmatrix}. \tag{9}$$

We will fix $J_1 = 1$ and vary $J_2$ in our studies.

**Experiments on inductive bias of $J_1$-$J_2$ Heisenberg Model.**

As shown previously, it can be challenging for neural networks to learn the sign structures. The ground state of the $J_1$-$J_2$ model also has a sign structure, which can be partially captured by the Marshall sign rule(Marshall, 1955). The sign rule is exact at $J_2 = 0$ and becomes worse for large $J_2$. This could introduce bias to the neural network wavefunction if directly applied. We compare the restricted Boltzmann machine (RBM) (from NetKet (Vicentini et al., 2022)), our implementation of gated PixelCNN, and the two different constructions of ANTN with and without the sign rule.

The result is shown in Table 1. Since our approach is based on the variational principle discussed in Sec. 3, it provides an upper bound on the exact ground state energy; therefore, a lower energy implies a better state. As shown in the results, both RBM and PixelCNN improve significantly with the application of the sign rule at $J_2 = 0.2$ and $J_2 = 0.5$, but the results deteriorate at $J_2 = 0.8$. This is expected because the sign rule becomes increasingly inaccurate as $J_2$ increases, especially past $J_2 = 0.5$. Our ANTN, on the other hand, does not require the sign rule in both constructions. As an additional comparison, we note that both of our ANTN constructions achieved better results compared to the recently developed matrix product backflow state (Lami et al., 2022), which uses the sign rule and has a per site energy of $-1.9267$ at $J_2 = 0.5$.

**Ablation study on $J_1$-$J_2$ Heisenberg Model.** We scan across many $J_2$ for the model without using the sign rule. In this experiment, we compare the performance of six models to compute the ground state energy of the $J_1$-$J_2$ Hamiltonian system with $J_1 = 1$ fixed and $J_2 = 0.2$-$0.8$ with $0.1$ increments, covering three different phases of the model. The first three models are TN models, using the MPS with bond dimensions of 8, 70, and 1024. For up to $4 \times 4$ system, the MPS results with bond dimension 1024 are exact and can be regarded as a benchmark. The fourth model is a PixelCNN pure neural network; the fifth and sixth models are elementwise and blockwise ANTN models. Thus the experiment compares the ANTN against the two previous state-of-the-art techniques.

Table 2 summarizes the per site ground state energy computed by different models at different $J_2$'s in three distinct phases. Our results provide strong evidence that ANTN integrating TN and ARNN outperforms each base model, achieving state-of-the-art results.

**Comparison between ANTN and MPS.** In all cases the ANTN surpasses the MPS with the corresponding bond dimension on which they are based. For example, even in the $10 \times 10$ system with $J_2 > 0.5$, where MPS with a bond dimension of 1024 outperforms the ANTN, the elementwise ANTN still significantly improves on the base MPS with a bond dimension of 8 and the blockwise ANTN improves on the base MPS with a bond dimension of 70. It is consistent with Theorem. 5.3 and Theorem. 5.2 that ANTN is more expressive than TN.

In addition, the ANTN models scale well for larger systems. Figure 3 visualizes the scalability of ANTN compared to MPS. The figure plots the difference in per site energy between the elementwise ANTN with bond dimension 8 and MPS with bond dimension 1024 for $J_2 = 0.2, 0.5, 0.8$ where lower energies signify better performance for the elementwise ANTN. Compared to MPS, the ANTN models compute better ground state energies as the system size grows larger. As the system size increases, the elementwise ANTN with bond dimension 8 starts to outperform MPS with bond dimension 1024. Even at $J_2 = 0.6$, where the elementwise ANTN is slightly worse than MPS, the difference gets significantly smaller at $12 \times 12$ compared to $10 \times 10$ (Table 2). In fact, the elementwise ANTN achieves such a performance using only $\sim 1\%$ the number of parameters of MPS. We note that for large $J_2$, the system goes into a stripped phase (Nomura & Imada, 2021), which can be less challenging for MPS to represent. Nevertheless, in almost all cases our ANTN still outperforms MPS on the $12 \times 12$ system. MPS has a cost dominated by the bond dimension (quadratic for memory and cubic for computational), which limits its use in practice for large system sizes that require large bond dimensions. According to the complexity analysis in Sec. 5.1, ANTN has lower complexity than TN and thus scales better for larger systems.

**Comparison between ANTN and ARNN.** The elementwise and blockwise ANTN models also consistently outperform the pure neural network PixelCNN model. This agrees with Theorem 5.3 that both elementwise and blockwise ANTN have a generalized expressivity compared to ARNN. In Appendix C we further compare ANTN with ARNN and find that: (a) it is challenging to surpass ANTN by ARNN even with an increased number of parameters; (b) the improvement from ANTN mainly comes from the effective inductive bias of MPS structure instead of the DMRG initialization; (c) ANTN has a favorable scaling compared to ARNN and MPS for reaching accurate results.

The details of all experiment setups and hyperparameters can be found in Appendix D.

# 7   Conclusion

In this paper, we developed Autoregressive Neural TensorNet, bridging the two state-of-the-art methods in the field, tensor networks, and autoregressive neural networks. We proved that Autoregressive Neural TensorNet is self-normalized with exact sampling, naturally generalizes the expressivity

of both tensor networks and autoregressive neural networks, and inherits proper physics inductive bias (e.g. sign structures) from tensor networks and various symmetries from autoregressive neural networks. We demonstrated our approach on quantum state learning and the challenging 2D $J_1$-$J_2$ Heisenberg model with different system sizes and couplings. Our approach achieves better performance than both the original tensor network and autoregressive neural network while surpassing tensor networks with large bond dimensions as the system size increases. In addition, our approach is robust, independent of sign rule. Besides scientific applications, since both tensor networks and autoregressive neural networks have been applied to machine learning tasks such as supervised learning and generative modeling, our novel approach holds promise for better performance in these domains due to its exact sampling, expressivity, and symmetries.

## Broader Impact

Our Autoregressive Neural TensorNet, blending tensor networks and autoregressive neural networks, could advance our grasp of quantum phenomena, potentially fueling scientific breakthroughs. It enhances quantum material design and quantum computing through improved simulations and control of quantum states. This technology may also inspire new machine learning models for handling high-dimensional data. However, possible ethical and societal impacts, such as the use for chemical weapon development, require careful scrutiny.

## Acknowledgement

The authors acknowledge support from the National Science Foundation under Cooperative Agreement PHY-2019786 (The NSF AI Institute for Artificial Intelligence and Fundamental Interactions, http://iaifi.org/). This material is based upon work supported by the U.S. Department of Energy, Office of Science, National Quantum Information Science Research Centers, Co-design Center for Quantum Advantage (C2QA) under contract number DE-SC0012704. This material is also in part based upon work supported by the Air Force Office of Scientific Research under the award number FA9550-21-1-0317. The authors acknowledge the MIT SuperCloud (Reuther et al., 2018) and Lincoln Laboratory Supercomputing Center for providing (HPC, database, consultation) resources that have contributed to the research results reported within this paper. Some computations in this paper were run on the FASRC cluster supported by the FAS Division of Science Research Computing Group at Harvard University. ZC acknowledges the DARPA 134371-5113608 award and NSF 10434.

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

# Appendices

## A  Additional Background

### A.1  Quantum State Learning

In this section, we discuss more details about quantum state learning. As mentioned in the main paper, the quantum fidelity

$$F = |\langle\phi|\psi_\theta\rangle|^2 = \left|\sum_{\boldsymbol{x}} \phi^*(\boldsymbol{x})\psi_\theta(\boldsymbol{x})\right|^2 \tag{A.1}$$

measures the closeness of the two (normalized) quantum states $|\phi\rangle$ and $|\psi_\theta\rangle$, with $^*$ denoting complex conjugation. By minimizing $-\log F$, one can obtain the $|\psi_\theta\rangle$ closest to the target state $|\phi\rangle$ (see Appendix D for optimization details). For small system sizes ($\lesssim 20$ qubits), it is possible to enumerate all the basis $\boldsymbol{x}$ and compute $-\log F$ exactly as in the case of this work. As mentioned in the main paper, we learn the random Bell state and shallow random circuit to test the expressivity and physics inductive bias of the ansatz. More specifically, the states are generated as follows

**Random Bell State.** A two-qubit Bell state is defined to be the state $(|00\rangle + |11\rangle)/\sqrt{2}$. We use the following algorithm to generate a $n$-qubit random Bell state ($n$ is a multiple of 2).

---
**Algorithm 1** Random Bell State Generation

---
$\boldsymbol{a} \leftarrow \text{shuffle}([1, \dots \frac{n}{2}])$
$\boldsymbol{b} \leftarrow \text{shuffle}([\frac{n}{2} + 1, \dots n])$
$\boldsymbol{\Psi} \leftarrow []$
$i \leftarrow 1$
**while** $i \leq n/2$ **do**
  $\boldsymbol{\Psi}.\text{append}(\text{bell\_state}(a_i, b_i))$
  $i \leftarrow i + 1$
**end while**
$|\psi\rangle \leftarrow \text{product\_state}(\boldsymbol{\Psi})$
**return** $|\psi\rangle$

---

where $\text{bell\_state}(\cdot, \cdot)$ creates a Bell state given the two qubits and $\text{product\_state}([])$ creates the tensor product state for a list of individual states. Each two-qubit Bell state in the random Bell state forms between the left half and right half of the system, allowing a maximum entanglement across a cut in the middle of the system.

**Shallow Random Circuit.** The shallow random circuit generates random states using the following algorithm.

---

**Algorithm 2** Shallow Random Circuit State Generation

---

$|\psi\rangle \leftarrow |0\rangle^{\otimes n}$
$l \leftarrow 1$
**while** $l \leq n_l$ **do**
    $i \leftarrow 1$
    **while** $i < n$ **do**
        $U \leftarrow \text{random\_gate}(i, i+1)$
        $|\psi\rangle \leftarrow U |\psi\rangle$
        $i \leftarrow i + 2$
    **end while**
    $i \leftarrow 2$
    **while** $i < n$ **do**
        $U \leftarrow \text{random\_gate}(i, i+1)$
        $|\psi\rangle \leftarrow U |\psi\rangle$
        $i \leftarrow i + 2$
    **end while**
**end while**
**return** $|\psi\rangle$

---

where $n_l$ is the number of layers, and $\text{random\_gate}(\cdot, \cdot)$ generates a random unitary gate on the two qubits. The $\text{random\_gate}(\cdot, \cdot)$ function is realized by first generating a (real-valued) Gaussian random matrix of size $4 \times 4$, following a QR decomposition of the matrix to obtain the unitary part as the random unitary gate.

Each of the above algorithms defines a distribution of states, which we average over multiple realizations to compute the mean and standard deviation of the learned quantum fidelity.

However, for large system sizes, the Hilbert space is too large, so one has to evaluate the fidelity stochastically. This can be achieved by rewriting

$$f := \sum_{\boldsymbol{x}} \phi^*(\boldsymbol{x}) \psi_\theta(\boldsymbol{x}) = \sum_{\boldsymbol{x}} |\phi(\boldsymbol{x})|^2 \frac{\psi_\theta(\boldsymbol{x})}{\phi(\boldsymbol{x})} = \mathbb{E}_{\boldsymbol{x} \sim |\phi|^2} \frac{\psi_\theta(\boldsymbol{x})}{\phi(\boldsymbol{x})}. \tag{A.2}$$

Which allows the fidelity to be evaluated by sampling from $|\phi|^2$. The gradient of $-\log F$ can be written as

$$\nabla_\theta - \log F = -2\Re \frac{\nabla_\theta f}{f} = -2\Re \frac{\mathbb{E}_{\boldsymbol{x} \sim |\phi|^2} \frac{\nabla_\theta \psi_\theta(\boldsymbol{x})}{\phi(\boldsymbol{x})}}{\mathbb{E}_{\boldsymbol{x} \sim |\phi|^2} \frac{\psi_\theta(\boldsymbol{x})}{\phi(\boldsymbol{x})}} = -2\Re \, \mathbb{E}_{\boldsymbol{x} \sim |\phi|^2} \frac{\frac{\psi_\theta(\boldsymbol{x})}{\phi(\boldsymbol{x})}}{\mathbb{E}_{\boldsymbol{x}' \sim |\phi|^2} \frac{\psi_\theta(\boldsymbol{x}')}{\phi(\boldsymbol{x}')}} \nabla_\theta \log \psi_\theta(\boldsymbol{x}). \tag{A.3}$$

Alternatively, one can sample from $\psi_\theta(\boldsymbol{x})$ by writing the complex conjugated

$$f^* = \sum_{\boldsymbol{x}} |\psi_\theta(\boldsymbol{x})|^2 \frac{\phi(\boldsymbol{x})}{\psi_\theta(\boldsymbol{x})} = \mathbb{E}_{\boldsymbol{x} \sim |\psi_\theta|^2} \frac{\phi(\boldsymbol{x})}{\psi_\theta(\boldsymbol{x})}. \tag{A.4}$$

This time, we have to be careful with

$$\nabla_\theta f^* = \sum_{\boldsymbol{x}} \phi(\boldsymbol{x}) \nabla_\theta \psi_\theta^*(\boldsymbol{x}) = \sum_{\boldsymbol{x}} |\psi_\theta(\boldsymbol{x})|^2 \frac{\phi(\boldsymbol{x})}{\psi_\theta(\boldsymbol{x})} \nabla_\theta \log \psi_\theta^*(\boldsymbol{x}) = \mathbb{E}_{\boldsymbol{x} \sim |\psi_\theta|^2} \frac{\phi(\boldsymbol{x})}{\psi_\theta(\boldsymbol{x})} \nabla_\theta \log \psi_\theta^*(\boldsymbol{x}). \tag{A.5}$$

Then,

$$\nabla_\theta - \log F = -2\Re \frac{\nabla_\theta f^*}{f^*} = -2\Re \, \mathbb{E}_{\boldsymbol{x} \sim |\psi_\theta|^2} \frac{\frac{\phi(\boldsymbol{x})}{\psi_\theta(\boldsymbol{x})}}{\mathbb{E}_{\boldsymbol{x}' \sim |\psi_\theta|^2} \frac{\phi(\boldsymbol{x}')}{\psi_\theta(\boldsymbol{x}')}} \nabla_\theta \log \psi_\theta^*(\boldsymbol{x}). \tag{A.6}$$

We can further use the variance reduction technique (Greensmith et al., 2004) from reinforcement learning and write

$$\nabla_\theta - \log F = -2\Re \, \mathbb{E}_{\boldsymbol{x} \sim |\psi_\theta|^2} \left[ \frac{\frac{\phi(\boldsymbol{x})}{\psi_\theta(\boldsymbol{x})}}{\mathbb{E}_{\boldsymbol{x}' \sim |\psi_\theta|^2} \frac{\phi(\boldsymbol{x}')}{\psi_\theta(\boldsymbol{x}')}} - 1 \right] \nabla_\theta \log \psi_\theta^*(\boldsymbol{x}), \tag{A.7}$$

where the minus 1 comes from the mean of the coefficient in front of $\nabla_\theta \log \psi_\theta^*(\boldsymbol{x})$

$$\frac{\mathbb{E}_{\boldsymbol{x}\sim|\psi_\theta|^2} \frac{\phi(\boldsymbol{x})}{\psi_\theta(\boldsymbol{x})}}{\mathbb{E}_{\boldsymbol{x}'\sim|\psi_\theta|^2} \frac{\phi(\boldsymbol{x}')}{\psi_\theta(\boldsymbol{x}')}} = 1. \tag{A.8}$$

One can verify that without the assumption that $|\psi_\theta\rangle$ is normalized (i.e. $F = |\langle\phi|\psi_\theta\rangle|^2 / \langle\psi_\theta|\psi_\theta\rangle$), Eq. A.7 can also be obtained without applying the variance reduction formula.

In the main paper, due to the small system size $N = 16$, we enumerated all the bases when evaluating the quantum fidelity.

## A.2 Variational Monte Carlo

As shown in the main paper, given a Hamiltonian $\hat{\mathcal{H}}$ (and its matrix elements $\mathcal{H}_{\boldsymbol{x},\boldsymbol{x}'}$)

$$\langle\psi_\theta|\hat{\mathcal{H}}|\psi_\theta\rangle = \sum_{\boldsymbol{x},\boldsymbol{x}'} \mathcal{H}_{\boldsymbol{x},\boldsymbol{x}'}\psi_\theta^*(\boldsymbol{x})\psi_\theta(\boldsymbol{x}') = \sum_{\boldsymbol{x},\boldsymbol{x}'} |\psi_\theta(\boldsymbol{x})|^2 \frac{\mathcal{H}_{\boldsymbol{x},\boldsymbol{x}'}\psi_\theta(\boldsymbol{x}')}{\psi_\theta(\boldsymbol{x})} = \mathbb{E}_{\boldsymbol{x}\sim|\psi_\theta|^2} \frac{\sum_{\boldsymbol{x}'} \mathcal{H}_{\boldsymbol{x},\boldsymbol{x}'}\psi_\theta(\boldsymbol{x}')}{\psi_\theta(\boldsymbol{x})}. \tag{A.9}$$

Since the Hamiltonian $\hat{\mathcal{H}}$ is usually sparse, give $\boldsymbol{x}$, one only needs to sum up a small number of $\boldsymbol{x}'$'s for the numerator (usually linear in system size), allowing Eq. A.9 to be evaluated efficiently. Analogously, we can evaluate the gradient as

$$\begin{aligned}
\nabla_\theta \langle\psi_\theta|\hat{\mathcal{H}}|\psi_\theta\rangle &= 2\Re \sum_{\boldsymbol{x},\boldsymbol{x}'} \mathcal{H}_{\boldsymbol{x},\boldsymbol{x}'} \left[\nabla_\theta\psi_\theta^*(\boldsymbol{x})\right]\psi_\theta(\boldsymbol{x}') \\
&= 2\Re \sum_{\boldsymbol{x},\boldsymbol{x}'} |\psi_\theta(\boldsymbol{x})|^2 \mathcal{H}_{\boldsymbol{x},\boldsymbol{x}'} \frac{\nabla_\theta\psi_\theta^*(\boldsymbol{x})}{\psi_\theta^*(\boldsymbol{x})} \frac{\psi_\theta(\boldsymbol{x}')}{\psi_\theta(\boldsymbol{x})} \\
&= 2\Re \mathbb{E}_{\boldsymbol{x}\sim|\psi_\theta|^2} \frac{\sum_{\boldsymbol{x}'} \mathcal{H}_{\boldsymbol{x},\boldsymbol{x}'}\psi_\theta(\boldsymbol{x}')}{\psi_\theta(\boldsymbol{x})} \nabla_\theta \log \psi_\theta^*(\boldsymbol{x}).
\end{aligned} \tag{A.10}$$

Furthermore, it is possible to reduce the variance by either directly applying the variance reduction formula (Greensmith et al., 2004), or explicitly normalizing $|\psi_\theta\rangle$ (minimizing $\langle\psi_\theta|\hat{\mathcal{H}}|\psi_\theta\rangle / \langle\psi_\theta|\psi_\theta\rangle$) to obtain

$$\nabla_\theta \langle\psi_\theta|\hat{\mathcal{H}}|\psi_\theta\rangle = 2\Re \mathbb{E}_{\boldsymbol{x}\sim|\psi_\theta|^2} \left[\frac{\sum_{\boldsymbol{x}'} \mathcal{H}_{\boldsymbol{x},\boldsymbol{x}'}\psi_\theta(\boldsymbol{x}')}{\psi_\theta(\boldsymbol{x})} - \langle\psi_\theta|\hat{\mathcal{H}}|\psi_\theta\rangle\right] \nabla_\theta \log \psi_\theta^*(\boldsymbol{x}), \tag{A.11}$$

where $\langle\psi_\theta|\hat{\mathcal{H}}|\psi_\theta\rangle$ can be approximated by applying Eq. A.9 on the same batch $\boldsymbol{x}$. In this work, we use Eq. A.11 as the gradient of the loss function (see Appendix D for optimization details).

# B Additional Theoretical Results

## B.1 Exact Sampling from Conditional Probabilities

Suppose a full probability distribution over multiple qubits is written as a product of conditional probabilities as

$$p(\boldsymbol{x}) = p(x_1, \ldots x_n) = \prod_{i=1}^{n} p(x_i|\boldsymbol{x}_{<i}) \tag{B.1}$$

with $\boldsymbol{x}_{<i} = (x_1, \ldots x_{i-1})$, then sampling a bitstring $\boldsymbol{x}$ from the probability distribution can be obtained by sequentially sampling each conditional probability as Since each sample is sampled independently using this algorithm, a batch of samples can be obtained in parallel, allowing for an efficient implementation on GPUs. In addition, the exact sampling nature of the algorithm makes it not suffer from the autocorrelation time problem of standard Markov chain Monte Carlo.

This same algorithm applies to sampling from $|\psi(\boldsymbol{x})|^2$ with a simple replacement $p(x_i|\boldsymbol{x}_{<i}) \rightarrow |\psi(x_i|\boldsymbol{x}_{<i})|^2$. This is a direct consequence of

$$\psi(\boldsymbol{x}) = \prod_{i=1}^{n} \psi(x_i|\boldsymbol{x}_{<i}) \Rightarrow |\psi(\boldsymbol{x})|^2 = \prod_{i=1}^{n} |\psi(x_i|\boldsymbol{x}_{<i})|^2, \tag{B.2}$$

and the fact that we require $\sum_{x_i} |\psi(x_i|\boldsymbol{x}_{<i})|^2 = 1$.

**Algorithm 3** Exact Sampling from Conditional Probabilities/Wavefunctions

---

$\boldsymbol{x} \leftarrow []$
$i \leftarrow 1$
**while** $i \leq n$ **do**
    $\boldsymbol{x}_{<i} \leftarrow \boldsymbol{x}$
    $x_i \sim p(x_i | \boldsymbol{x}_{<i})$
    $\boldsymbol{x}$.append$(x_i)$
    $i \leftarrow i + 1$
**end while**
**return** $\boldsymbol{x}$

---

## B.2 Exact Sampling from MPS

As mentioned in the main text, matrix product state (MPS) with particular constraints, also allows for exact sampling. Recall that MPS defines a wavefunction as

$$\psi(\boldsymbol{x}) = \sum_{\alpha_1, \ldots, \alpha_{n-1}} M_{x_1}^{\alpha_0 \alpha_1} \cdots M_{x_n}^{\alpha_{n-1} \alpha_n} = \sum_{\boldsymbol{\alpha}} \prod_{i=1}^{n} M_{x_i}^{\alpha_{i-1} \alpha_i}. \tag{B.3}$$

Let's first assume that the wavefunction is normalized. It is easy to notice that any gauge transformation

$$(M_{x_i}, M_{x_{i+1}}) \rightarrow (M_{x_i} A, A^{-1} M_{x_{i+1}}), \tag{B.4}$$

with $A$ being an invertible matrix leaves the resulting wavefunction invariant. Here, we suppress the $\alpha$ indices and view them as the indices for matrices. It has been shown that by fixing this gauge redundancy (Vidal, 2003, 2004), we can restrict all the $M$ matrices to satisfy the following condition

$$\sum_{x_i, \alpha_i} M_{x_i}^{\alpha_{i-1} \alpha_i} \left( M_{x_i}^{\alpha'_{i-1} \alpha_i} \right)^* = \delta_{\alpha_{i-1}, \alpha'_{i-1}}, \tag{B.5}$$

where $\delta_{\cdot, \cdot}$ is the Kronecker delta function. The matrices $M$ that satisfy this condition are called isometries, and the MPS is called in the right canonical form (to be distinguished from the left canonical form). In the right canonical form, each $M_{x_i}^{\alpha_{i-1} \alpha_i}$ can be interpreted as a basis transformation $(x_i, \alpha_i) \rightarrow \alpha_{i-1}$ that is part of a unitary matrix.

**Theorem B.1.** *Defining the left partially contracted tensors*

$$M_{L \boldsymbol{x}_{\leq j}}^{\alpha_j} := \sum_{\boldsymbol{\alpha}_{\leq j}} \prod_{i=1}^{j} M_{x_i}^{\alpha_{i-1} \alpha_i}, \tag{B.6}$$

*If the matrix product state is in the right canonical form, then the marginal probability of the wavefunction satisfies*

$$q(\boldsymbol{x}_{\leq i}) := \sum_{\boldsymbol{x}_{>i}} |\psi(\boldsymbol{x})|^2 = \sum_{\alpha_j} M_{L \boldsymbol{x}_{\leq j}}^{\alpha_j} \left( M_{L \boldsymbol{x}_{\leq j}}^{\alpha_j} \right)^*. \tag{B.7}$$

*Proof.* Let's define the right partially contracted tensors analogously

$$M_{R \boldsymbol{x}_{\geq j}}^{\alpha_{j-1}} := \sum_{\boldsymbol{\alpha}_{\geq j}} \prod_{i=j}^{n} M_{x_i}^{\alpha_{i-1} \alpha_i}. \tag{B.8}$$

We will first show that

$$\sum_{\boldsymbol{x}_{\geq j}} M_{R \boldsymbol{x}_{\geq j}}^{\alpha_{j-1}} \left( M_{R \boldsymbol{x}_{\geq j}}^{\alpha'_{j-1}} \right)^* = \delta_{\alpha_{j-1}, \alpha'_{j-1}}. \tag{B.9}$$

We can prove this by induction:

- **Base case:**

$$\sum_{x_n} M_{R x_n}^{\alpha_{n-1}} \left( M_{R x_n}^{\alpha_{n-1}} \right)^* = \sum_{x_n} M_{x_n}^{\alpha_{n-1} \alpha_n} \left( M_{x_n}^{\alpha'_{n-1} \alpha'_n} \right)^* = \delta_{\alpha_{n-1}, \alpha'_{n-1}}, \tag{B.10}$$

    which directly comes from Eq. B.5 by noticing that $\mathcal{D}(\alpha_n) = 1$ so it can be ignored.

- **Inductive step:** Write

$$M_{R\boldsymbol{x}_{\geq j}}^{\alpha_{j-1}} = \sum_{\alpha_j} M_{x_j}^{\alpha_{j-1}\alpha_j} M_{R\boldsymbol{x}_{\geq j+1}}^{\alpha_j}. \tag{B.11}$$

Assuming

$$\sum_{\boldsymbol{x}_{\geq j+1}} M_{R\boldsymbol{x}_{\geq j+1}}^{\alpha_j} \left( M_{R\boldsymbol{x}_{\geq j+1}}^{\alpha'_j} \right)^* = \delta_{\alpha_j, \alpha'_j}, \tag{B.12}$$

then

$$\begin{aligned}
\sum_{\boldsymbol{x}_{\geq j}} M_{R\boldsymbol{x}_{\geq j}}^{\alpha_{j-1}} \left( M_{R\boldsymbol{x}_{\geq j}}^{\alpha'_{j-1}} \right)^* &= \sum_{x_j} \sum_{\alpha_j, \alpha'_j} M_{x_j}^{\alpha_{j-1}\alpha_j} \left( M_{x_j}^{\alpha'_{j-1}\alpha'_j} \right)^* \sum_{\boldsymbol{x}_{\geq j+1}} M_{R\boldsymbol{x}_{\geq j+1}}^{\alpha_j} \left( M_{R\boldsymbol{x}_{\geq j+1}}^{\alpha'_j} \right)^* \\
&= \sum_{x_j} \sum_{\alpha_j, \alpha'_j} M_{x_j}^{\alpha_{j-1}\alpha_j} \left( M_{x_j}^{\alpha'_{j-1}\alpha'_j} \right)^* \delta_{\alpha_j, \alpha'_j} \\
&= \sum_{x_j, \alpha_j} M_{x_j}^{\alpha_{j-1}\alpha_j} \left( M_{x_j}^{\alpha'_{j-1}\alpha_j} \right)^* \\
&= \delta_{\alpha_{j-1}, \alpha'_{j-1}}.
\end{aligned} \tag{B.13}$$

Using this result, we can then prove our desired result about the marginal probability. Let's write

$$\psi(\boldsymbol{x}) = \sum_{\boldsymbol{\alpha}} \prod_{i=1}^n M_{x_i}^{\alpha_{i-1}\alpha_i} = \sum_{\alpha_j} M_{L\boldsymbol{x}_{\leq j}}^{\alpha_j} M_{R\boldsymbol{x}_{>j}}^{\alpha_j}, \tag{B.14}$$

then,

$$\begin{aligned}
q(\boldsymbol{x}_{\leq i}) = \sum_{\boldsymbol{x}_{>i}} |\psi(\boldsymbol{x})|^2 &= \sum_{\boldsymbol{x}_{>j}} \sum_{\alpha_j, \alpha'_j} M_{L\boldsymbol{x}_{\leq j}}^{\alpha_j} M_{R\boldsymbol{x}_{>j}}^{\alpha_j} \left( M_{L\boldsymbol{x}_{\leq j}}^{\alpha'_j} M_{R\boldsymbol{x}_{>j}}^{\alpha'_j} \right)^* \\
&= \sum_{\alpha_j, \alpha'_j} M_{L\boldsymbol{x}_{\leq j}}^{\alpha_j} \left( M_{L\boldsymbol{x}_{\leq j}}^{\alpha'_j} \right)^* \delta_{\alpha_j, \alpha'_j} \\
&= \sum_{\alpha_j} M_{L\boldsymbol{x}_{\leq j}}^{\alpha_j} \left( M_{L\boldsymbol{x}_{\leq j}}^{\alpha_j} \right)^*.
\end{aligned} \tag{B.15}$$

$\square$

**Corollary B.1.1.** *Matrix product state allows for efficient exact sampling.*

*Proof.* Thm. B.1 shows that marginal probability distribution $q(\boldsymbol{x}_{\leq i})$ can be evaluated efficiently. The conditional probability can be evaluated efficiently either as $q(x_i|\boldsymbol{x}_{<i}) = q(\boldsymbol{x}_{\leq i})/q(\boldsymbol{x}_{<i})$ or (equivalently) as an explicit normalization of the marginal probability distribution $q(x_i|\boldsymbol{x}_{<i}) = q(\boldsymbol{x}_{\leq i})/\sum_{x_i} q(\boldsymbol{x}_{\leq i})$. Then, Algorithm 3 can be used to sample from the full distribution. $\square$

### B.3 Autoregressive Sampling Order

Algorithm 3 samples from conditional probabilities, which requires an ordering of the system to be defined. For 1D systems, we use a linear ordering such that qubit 1 corresponds to the leftmost qubit, and qubit $n$ corresponds to the rightmost qubit. This ordering is natural for 1D MPS and transformer, as well as the transformer-based ANTN. For 2D systems, we use a snake (zig-zag) ordering. In this ordering, the qubit at the 2D location $(i, j)$ is defined to be the $i \times L_y + j$ th qubit, where $L_y$ is the number of qubits along the second dimension. This ordering is inherently defined from the masked convolution for PixelCNN and PixelCNN-based ANTN. The MPS is reshaped to satisfy the same ordering for 2D systems. The same 2D ordering can be generalized to other tensor networks as well.

### B.4 Additional Details of Expressivity and Symmetry Results of ANTN

**Theorem B.2.** *Both tensor networks and autoregressive neural networks are special cases of Autoregressive Neural TensorNet.*

*Proof.* Recall that the ANTN defines the wavefunction from its amplitude and phase

$$\psi(\boldsymbol{x}) := \sqrt{q(\boldsymbol{x})}e^{i\phi(\boldsymbol{x})}, \tag{B.16}$$

where

$$q(\boldsymbol{x}) = \prod_{j=1}^{n} q(x_j|\boldsymbol{x}_{<j}) \tag{B.17}$$

$$\text{with} \quad q(x_j|\boldsymbol{x}_{<j}) = \frac{q(\boldsymbol{x}_{\leq j})}{\sum_{x_j} q(\boldsymbol{x}_{\leq j})} \tag{B.18}$$

$$q(\boldsymbol{x}_{\leq j}) := \sum_{\alpha_j} \tilde{\psi}_L^{\alpha_j}(\boldsymbol{x}_{\leq j})\tilde{\psi}_L^{\alpha_j}(\boldsymbol{x}_{\leq j})^* \tag{B.19}$$

$$\tilde{\psi}_L^{\alpha_j}(\boldsymbol{x}_{\leq j}) := \sum_{\boldsymbol{\alpha}_{<j}} \prod_{i=1}^{j} \tilde{\psi}^{\alpha_{i-1}\alpha_i}(x_i|\boldsymbol{x}_{<i}) \tag{B.20}$$

and

$$\phi(\boldsymbol{x}) =: \operatorname{Arg} \sum_{\boldsymbol{\alpha}} \prod_{i=1}^{n} \tilde{\psi}^{\alpha_{i-1},\alpha_i}(x_i|\boldsymbol{x}_{<i}). \tag{B.21}$$

- **ANTN reduces to MPS.** If we don't modify the tensors with neural networks, $\tilde{\psi}^{\alpha_{i-1},\alpha_i}(x_i|\boldsymbol{x}_{<i})$ reduces to $M_{x_i}^{\alpha_{i-1},\alpha_i}$ and $\tilde{\psi}_L^{\alpha_j}(\boldsymbol{x}_{\leq j})$ reduces to $M_{L\boldsymbol{x}_{\leq j}}^{\alpha_j}$. It is then straightforward that Eq. B.19 reduces to Eq. B.15 and thus the probability distribution of ANTN reduces to that of MPS. Analogously, Eq. B.21 reduces to the phase of an MPS, and thus ANTN wavefunction reduces to an MPS wavefunction.

- **ANTN reduces to ARNN.** If we set all $\mathcal{D}(\alpha) = 1$, then $\tilde{\psi}^{\alpha_{i-1}\alpha_i}(x_i|\boldsymbol{x}_{<i})$ reduces to $\psi(x_i|\boldsymbol{x}_{<i})$ and all the summations over $\boldsymbol{\alpha}$ can be dropped. In this case, $q(\boldsymbol{x}_{\leq j})$ reduces to $\prod_{i=1}^{j} |\psi(x_i|\boldsymbol{x}_{<i})|^2$ and $q(x_j|\boldsymbol{x}_{<j})$ reduces to $|\psi(x_j|\boldsymbol{x}_{<j})|^2$, which is the same as the standard ARNN. In addition, the phase $\phi(\boldsymbol{x})$ reduces to $\operatorname{Arg} \prod_{i=1}^{n} \psi(x_i|\boldsymbol{x}_{<i})$ which is also the same as ARNN. Thus, ANTN wavefunction reduces to ARNN wavefunction.

Notice that we only showed one type of TN, i.e. MPS, but the proof can be easily generalized to any TN that allows evaluation of marginal probabilities. $\qquad\square$

**Theorem B.3.** *Autoregressive Neural TensorNet can be written as either a tensor network or an autoregressive neural network with an exponential (in system size) number of parameters.*

*Proof.*

- **ANTN as TN.** In ANTN, the base ARNN outputs the conditional tensors $\tilde{\psi}^{\alpha_{i-1}\alpha_i}(x_i|\boldsymbol{x}_{<i})$ to form the final wavefunctions. These conditional tensors can be written as $\tilde{\psi}_{x_1,x_2\dots,x_i}^{\alpha_{i-1}\alpha_i}$ in tensor notation. It directly follows from the shape of these tensors that they contain $\chi^2 \cdot 2^i$ elements (where $\chi^2$ is from the two bond dimensions and $2^i$ is from all the qubits at or before the current qubit $i$). According to Theorem 5.2 ARNN has volume law entanglement while TN doesn't. The conditional tensors generated from ARNN do not permit efficient tensor decomposition. Therefore, a tensor network almost has to parameterize all full conditional tensors, resulting in $\sum_{i=1}^{N} \chi^2 \cdot 2^i \sim \mathcal{O}(\chi^2 \cdot 2^N)$ parameters in total.

- **ANTN as ARNN.** In ANTN, the conditional probability is calculated from the marginal probability as shown in Eq. 4 as

$$q(\boldsymbol{x}_{\leq i}) = \sum_{\alpha_i} \tilde{\psi}_L^{\alpha_i}(\boldsymbol{x}_{\leq i})\tilde{\psi}_L^{\alpha_i}(\boldsymbol{x}_{\leq i})^*$$

$$= \sum_{\alpha_1,\alpha_1',\dots,\alpha_{i-1},\alpha_{i-1}'\alpha_i} \tilde{\psi}^{\alpha_1}(x_1)\tilde{\psi}^{\alpha_1'}(x_1)^* \cdots \tilde{\psi}^{\alpha_{i-1}\alpha_i}(x_i|\boldsymbol{x}_{<i})\tilde{\psi}^{\alpha_{i-1}'\alpha_i}(x_i|\boldsymbol{x}_{<i})^*. \tag{B.22}$$

The summation over $\alpha_i$'s contains $\chi^{2i-1}$ terms, each of which can be viewed as a marginal (quasi-)probability $q^{\alpha_1,\alpha_1',\dots,\alpha_i}(\boldsymbol{x}_{\leq i})$ generated by the underlying ARNN of ANTN. This

effectively creates a weight matrix on top of a conventional ARNN, whose shape is $h_{\dim} \times \chi^{2i-1} \times 2$ (where $h_{\dim}$ is the hidden dimension, and 2 is the local dimension of each qubit) on top of a conventional ARNN, resulting in $\sum_{i=1}^{N} 2 \cdot h_{\dim} \cdot \chi^{2i-1} \sim \mathcal{O}(h_{\dim} \cdot \chi^{2N})$ parameters.

$\square$

**Definition B.1** (Mask Symmetry). A conditional wavefunction $\psi(x_i|\boldsymbol{x}_{<i})$ has a *mask symmetry* if $\psi(x_i|\boldsymbol{x}_{<i}) = 0$ for some $x_i$ given $\boldsymbol{x}_{<i}$.

**Theorem B.4.** *Autoregressive Neural TensorNet inherits mask symmetry from autoregressive neural network.*

*Proof.* The mask symmetry results in $\psi(\boldsymbol{x}) = 0$ for certain $\boldsymbol{x}$ satisfying the condition while preserving the normalization of $\psi(\boldsymbol{x})$. For ANTN, we can directly set $q(x_i|\boldsymbol{x}_{<i}) = 0$ for the $x_i$ given $\boldsymbol{x}_{<i}$. This results in $\psi(\boldsymbol{x}) = 0$ for the same $\boldsymbol{x}$. $\square$

**Corollary B.4.1** (Global U(1) Symmetry). *Autoregressive Neural TensorNet can realize global U(1) symmetry, which conserves particle number.*

*Proof.* A global U(1) symmetry in a qubit system manifests as a conservation of the number of 0's and number of 1's in $\boldsymbol{x}$. Equivalently, $\psi(\boldsymbol{x}) = 0$ for $\boldsymbol{x}$ violates such conservation. (Hibat-Allah et al., 2020) has shown that autoregressive neural networks can preserve global U(1) symmetry as a mask symmetry, which ANTN inherits. $\square$

**Definition B.2** (Function Symmetry). A conditional wavefunction tensor $\psi(x_i|\boldsymbol{x}_{<i})$ has a *function symmetry* over a function $F$ if $\psi(x_i|\boldsymbol{x}_{<i}) = \psi(F(x_i; \boldsymbol{x}_{<i})|\boldsymbol{F}(\boldsymbol{x}_{<i}))$ where $\boldsymbol{F}(\boldsymbol{x}_{\leq i}) := \{F(x_1), F(x_2; x_1), \ldots, F(x_i; \boldsymbol{x}_{<i})\}$.

**Theorem B.5.** *Autoregressive Neural TensorNet inherits function symmetry from autoregressive neural networks.*

*Proof.* We can apply the function symmetry on the left partially contracted tensors instead of the conditional wavefunctions. Here, we show that this produces the desired conditional probabilities and phase. Applying the function symmetry on the left partially contracted tensors results in

$$\tilde{\psi}_L^{\alpha_j}(\boldsymbol{x}_{\leq j}) = \tilde{\psi}_L^{\alpha_j}(F(\boldsymbol{x}_{\leq j})). \tag{B.23}$$

This implies that the marginal probability satisfies $q(\boldsymbol{x}_{\leq j}) = q(F(\boldsymbol{x}_{\leq j}))$ after contracting over the index $\alpha_j$, which then results in the correct symmetry on the conditional probabilities

$$q(x_j|\boldsymbol{x}_{<j}) = q(F(x_j)|F(\boldsymbol{x}_{<j})), \tag{B.24}$$

as the conditional probability is just the marginal probability normalized at site $j$. The phase, on the other hand,

$$\phi(\boldsymbol{x}) = \text{Arg} \sum_{\boldsymbol{\alpha}} \prod_{i=1}^{n} \tilde{\psi}^{\alpha_{i-1}, \alpha_i}(x_i|\boldsymbol{x}_{<i}) = \text{Arg}\, \tilde{\psi}_L^{\alpha_n}(\boldsymbol{x}_{\leq n}), \tag{B.25}$$

which satisfies the same function symmetry from Eq. B.23. $\square$

**Corollary B.5.1** ($\mathbb{Z}_2$ Spin Flip Symmetry). *Autoregressive Neural TensorNet can realize $\mathbb{Z}_2$ spin flip symmetry such that the ANTN wavefunction is invariant under conjugation of the input.*

*Proof.* Spin flip symmetry exists for quantum chemistry systems that do not couple spin up and spin down electron, so that the wavefunction is invariant to inputs when spin up and down are flipped. (Barrett et al., 2022) has shown that autoregressive neural network can preserve $\mathbb{Z}_2$ spin flip symmetry as a function symmetry, which ANTN inherits. $\square$

**Corollary B.5.2** (Discrete Abelian and Non-Abelian Symmetries). *Autoregressive Neural TensorNet can realize discrete Abelian and Non-Abelian symmetries.*

*Proof.* Gauge symmetry is a local symmetry such that the function is invariant under local transformation. (Luo et al., 2021b) has shown that autoregressive neural networks can preserve discrete Abelian and non-Abelian symmetries as either the mask symmetry or the function symmetry, which ANTN inherits. $\square$

## C Additional Experiments

In this section, we perform additional experiments to further compare ANTN and ARNN, and show that

- It is challenging to surpass ANTN by ARNN even with increased number of parameters.
- The improvement from ANTN mainly comes from the effective inductive bias of MPS structure instead of the DMRG initialization.
- ANTN has a favorable scaling compared to ARNN and MPS for reaching accurate results.

| Energy per site ↓ and additional information for $10 \times 10$ system | | | | | | | |
|---|---|---|---|---|---|---|---|
| Algorithms | $J_2 = 0.2$ | $J_2 = 0.5$ | $J_2 = 0.8$ | $N_{\text{params,TN}}$ | $N_{\text{params,NN}}$ | $N_{\text{params}}$ | $T_{\text{training}}$ |
| MPS (8) | -1.997537 | -1.734326 | -1.869871 | $1.2 \times 10^4$ | 0 | $1.2 \times 10^4$ | < 1 hr |
| MPS (70) | -2.191048 | -1.866159 | -2.007845 | $8.8 \times 10^5$ | 0 | $8.8 \times 10^5$ | < 1 hr |
| MPS (1024) | -2.255633 | -1.938770 | **-2.062730** | $1.7 \times 10^8$ | 0 | $1.7 \times 10^8$ | 242 hrs |
| PixelCNN (7-48) | -2.22462(24) | -1.74098(29) | -1.98331(17) | 0 | $5.0 \times 10^5$ | $5.0 \times 10^5$ | 43 hrs |
| **PixelCNN (P) (7-48)** | -2.1920(9) | -1.6458(10) | -1.9468(6) | 0 | $5.0 \times 10^5$ | $5.0 \times 10^5$ | 19 hrs$^†$ |
| **PixelCNN (8-48)** | -2.23581(14) | -1.86922(16) | -2.01093(16) | 0 | $5.7 \times 10^5$ | $5.7 \times 10^5$ | 59 hrs |
| EW (7-48-8) | **-2.26034(6)** | -1.94001(6) | -2.05707(43) | $1.2 \times 10^6$ | $5.0 \times 10^5$ | $1.7 \times 10^6$ | 89 hrs |
| **EW (RI) (7-48-8)** | -2.25946(6) | **-1.94030(9)** | -2.05125(11) | $1.2 \times 10^6$ | $5.0 \times 10^5$ | $1.7 \times 10^6$ | 46 hrs$^†$ |
| BW (7-48-70) | -2.25755(8) | -1.93842(42) | -2.05088(14) | $8.9 \times 10^5$ | $5.0 \times 10^5$ | $1.4 \times 10^6$ | 136 hrs |

Table 3: Energy per site ↓, tensor network parameters $N_{\text{params,TN}}$, neural network parameters $N_{\text{params,NN}}$, total parameters $N_{\text{params}}$ and runtime $T_{\text{training}}$ for $10 \times 10$ system with various algorithms. Elementwise (EW) and blockwise (BW) are two constructions of ANTN with PixelCNN as the underlying ARNN. For MPS, the number in the parenthesis labels the bond dimension. For PixelCNN, the two numbers label the number of layers and hidden dimension respectively. For ANTN, the numbers label (in this order) the number of layers, hidden dimension, and bond dimension. For PixelCNN, we also test pertaining (P) using MPS result (with bond dimension the last number in parenthesis), and for EW, we in addition test random initialization (RI) of the MPS part. The algorithms in boldface are new experiments not previously shown in the main paper. Best energies for each $J_2$ are also highlighted in boldface. The MPS with DMRG algorithm is run on CPUs, while the PixelCNN, elementwise (EW) and blockwise (BW) ANTN are trained using GPUs, with PixelCNN (P) (7-48) and EW (RI) (7-48-8) using A100 GPUs (labeled with $^†$), and the rest using V100 GPUs.

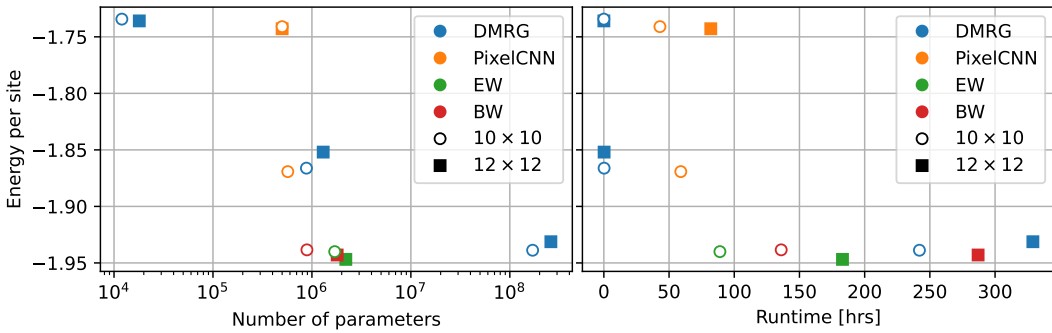

Figure 4: Energy per site vs total number of parameters and runtime (in hours) for various algorithms (MPS with DMRG algorithm, PixelCNN, elementwise (EW), and blockwise (BW) ANTNs) and system sizes ($10 \times 10$ and $12 \times 12$) for $J_2 = 0.5$. The ANTN construction uses PixelCNN as the underlying ARNN. The total number of parameters includes parameters from both the TN part and the ARNN part.

### C.1 Comparison between ANTN and ARNN with More Layers

In Table 3 and 4 we include additional results.

| Energy per site ↓ and additional information for $10 \times 10$ system at $J_2 = 0.5$ | | | | | |
|---|---|---|---|---|---|
| Algorithms | $J_2 = 0.5$ | $N_{\mathrm{params,TN}}$ | $N_{\mathrm{params,NN}}$ | $N_{\mathrm{params}}$ | $T_{\mathrm{training}}$ |
| MPS (8) | -1.734326 | $1.2 \times 10^4$ | 0 | $1.2 \times 10^4$ | < 1 hr |
| MPS (70) | -1.866159 | $8.8 \times 10^5$ | 0 | $8.8 \times 10^5$ | < 1 hr |
| MPS (1024) | -1.938770 | $1.7 \times 10^8$ | 0 | $1.7 \times 10^8$ | 242 hrs |
| PixelCNN (7-48) | -1.74098(29) | 0 | $5.0 \times 10^5$ | $5.0 \times 10^5$ | 19 hrs (A100) or 43 hrs (V100) |
| **PixelCNN (8-48)** | -1.86922(16) | 0 | $5.7 \times 10^5$ | $5.7 \times 10^5$ | 59 hrs (V100) |
| **PixelCNN (9-48)** | -1.90021(15) | 0 | $6.3 \times 10^5$ | $6.3 \times 10^5$ | 21 hrs (A100) |
| **PixelCNN (10-48)** | -1.90440(14) | 0 | $7.0 \times 10^5$ | $7.0 \times 10^5$ | 23 hrs (A100) |
| **PixelCNN (11-48)** | -1.92826(11) | 0 | $7.7 \times 10^5$ | $7.7 \times 10^5$ | 26 hrs (A100) |
| **PixelCNN (12-48)** | -1.92986(10) | 0 | $8.4 \times 10^5$ | $8.4 \times 10^5$ | 30 hrs (A100) |
| **PixelCNN (15-48)** | -1.92869(11) | 0 | $1.0 \times 10^6$ | $1.0 \times 10^6$ | 51 hrs (A100) |
| **PixelCNN (20-48)** | -1.91537(13) | 0 | $1.4 \times 10^6$ | $1.4 \times 10^6$ | 66 hrs (A100) |
| **PixelCNN (8-72)** | -1.88536(19) | 0 | $1.3 \times 10^6$ | $1.3 \times 10^6$ | 35 hrs (A100) |
| **PixelCNN (9-72)** | -1.89266(15) | 0 | $1.4 \times 10^6$ | $1.4 \times 10^6$ | 40 hrs (A100) |
| **PixelCNN (10-72)** | -1.90718(15) | 0 | $1.6 \times 10^6$ | $1.6 \times 10^6$ | 47 hrs (A100) |
| **PixelCNN (11-72)** | -1.92965(13) | 0 | $1.7 \times 10^6$ | $1.7 \times 10^6$ | 52 hrs (A100) |
| **PixelCNN (12-72)** | -1.91186(14) | 0 | $1.9 \times 10^6$ | $1.9 \times 10^6$ | 57 hrs (A100) |
| **PixelCNN (S) (11-48)** | -1.92915(12) | 0 | $7.7 \times 10^5$ | $7.7 \times 10^5$ | 31 hrs (A100) |
| **PixelCNN (S) (12-48)** | -1.93112(11) | 0 | $8.4 \times 10^5$ | $8.4 \times 10^5$ | 34 hrs (A100) |
| **PixelCNN (S) (15-48)** | -1.93233(9) | 0 | $1.0 \times 10^6$ | $1.0 \times 10^6$ | 47 hrs (A100) |
| **PixelCNN (S) (20-48)** | -1.93058(10) | 0 | $1.4 \times 10^6$ | $1.4 \times 10^6$ | 64 hrs (A100) |
| **PixelCNN (S) (11-72)** | -1.93211(10) | 0 | $1.7 \times 10^6$ | $1.7 \times 10^6$ | 54 hrs (A100) |
| **PixelCNN (S) (12-72)** | -1.92812(12) | 0 | $1.9 \times 10^6$ | $1.9 \times 10^6$ | 59 hrs (A100) |
| EW (7-48-8) | **-1.94001(6)** | $1.2 \times 10^6$ | $5.0 \times 10^5$ | $1.7 \times 10^6$ | 46 hrs (A100) or 89 hrs (V100) |
| BW (7-48-70) | -1.93842(42) | $8.9 \times 10^5$ | $5.0 \times 10^5$ | $1.4 \times 10^6$ | 136 hrs (V100) |

Table 4: Energy per site ↓, tensor network parameters $N_{\mathrm{params,TN}}$, neural network parameters $N_{\mathrm{params,NN}}$, total parameters $N_{\mathrm{params}}$ and runtime $T_{\mathrm{training}}$ for $10 \times 10$ system at $J_2 = 0.5$ with various algorithms. Elementwise (EW) and blockwise (BW) are two constructions of ANTN with PixelCNN as the underlying ARNN. For MPS, the number in the parenthesis labels the bond dimension. For PixelCNN, the two numbers label the number of layers and hidden dimension respectively. For ANTN, the numbers label (in this order) the number of layers, hidden dimension, and bond dimension. For PixelCNN, we also test the sign rule (S) The algorithms in boldface are new experiments not previously shown in the main paper. Best energies for each $J_2$ are also highlighted in boldface. The MPS with DMRG algorithm is run on CPUs, while the PixelCNN, elementwise (EW) and blockwise (BW) ANTN are trained using GPUs.

| Additional information for $12 \times 12$ system | | | | |
|---|---|---|---|---|
| Algorithms | $N_{\mathrm{params,TN}}$ | $N_{\mathrm{params,NN}}$ | $N_{\mathrm{params}}$ | $T_{\mathrm{training}}$ |
| MPS (8) | $1.8 \times 10^4$ | 0 | $1.8 \times 10^4$ | < 1 hr |
| MPS (70) | $1.3 \times 10^6$ | 0 | $1.3 \times 10^6$ | < 1 hr |
| MPS (1024) | $2.6 \times 10^8$ | 0 | $2.6 \times 10^8$ | 329 hrs |
| PixelCNN (7-48) | 0 | $5.0 \times 10^5$ | $5.0 \times 10^5$ | 82 hrs |
| EW (7-48-8) | $1.7 \times 10^6$ | $5.0 \times 10^5$ | $2.2 \times 10^6$ | 183 hrs |
| BW (7-48-70) | $1.3 \times 10^6$ | $5.0 \times 10^5$ | $1.8 \times 10^6$ | 287 hrs |

Table 5: Additional information for the number of parameters and runtime of the $12 \times 12$ experiment in the paper for various algorithms. The runtime is average over different $J_2$'s. The MPS with DMRG algorithm is run on CPUs, while the PixelCNN, elementwise (EW), and blockwise (BW) ANTN are trained using V100 GPUs. The ANTN uses PixelCNN as the underlying ARNN. For MPS, the number in the parenthesis labels the bond dimension. For PixelCNN, the two numbers label the number of layers and hidden dimension respectively. For ANTN, the numbers label (in this order) the number of layers, hidden dimension, and bond dimension.

In Table 3, we perform 2 addition tests: a) pretrain pure PixelCNN by learning MPS results; b) train ANTN (elementwise) without DMRG initialization. We find that DMRG pretraining negatively affects the result, while ANTN without DMRG initialization performs as good as with DMRG initialization. This is understandable, as learning MPS results with complex sign structure is essentially the same task as learning a quantum state from shallow random circuit, which has been shown a difficult task for ARNN in Figure 2(b) of the original paper. Since the MPS result is an approximation of the actual ground state, learning this state actually makes a worse initialization for PixelCNN. (We note that the PixelCNN was originally initialized from the result of smaller system sizes, as we described in Appendix D under transfer learning.) On the other hand, the ANTN, despite not using DMRG initialization, still retains the ability to represent flexible sign structures. In addition, since DMRG is just an optimization algorithm to train MPS, the gradient-descent-based algorithm should still train the MPS part of ANTN reasonably well without DMRG initialization.

In Table 4, we test a series of ARNNs with different layers (up to 20 layers) and hidden dimensions (up to 72 hidden dimensions) such that the largest ARNN tested has more parameters than ANTN. We found that all the ARNN energies are not as good at ANTN (which is only based on 7 layers and 48 hidden dimensions), despite that ARNNs have more parameters and take a longer time to train using the same GPU (see new data below). Furthermore, we observe that an increase of parameters can help ARNN to improve energy in general, but this improvement hits a diminishing return potentially due to difficulties in optimization as the number of parameters increases. In addition, we further tested the (approximate) sign rule and found that it could help ARNN obtain better energies but still worse than ANTN without the sign rule. This indicates that the ANTN has the advantage of inheriting the flexible sign structure from MPS, avoiding the manual bias of adding the sign rule. The new results have provided strong support for our expectation, both from an expressivity perspective and from the fact that ANTN has a better physics inductive bias than ARNN for optimization. We have also added a plot for the new benchmark in the updated manuscript. Although the added layer improves the result to some extent, the PixelCNN fails to beat ANTN in terms of energy calculation. This is consistent with Theorem 5.3 that ANTN is more expressive than ARNN.

### C.2 Comparison of Runtime and Number of Parameters

In Table 3, Table 4, Table 5 and Figure 4, we show the number of parameters and runtime of each algorithm. In summary, MPS has the most parameters, and is the slowest with DMRG algorithm, taking as long as 2 weeks. The ANTN is much more efficient compared to MPS while obtaining better energies, and it is comparable to pure ARNN while obtaining much better energies. Within the two types of ANTN, the elementwise construction, despite using more parameters, runs faster than the blockwise construction. We note that since TNs and ARNNs use the parameters very differently, a direct comparison of the two types of parameters may be unfair. Therefore, we list both the total number of parameters and the parameters for the individual parts (TN part and ARNN part).

## D   Experimental Details and Reproducibility

We note that while the following details can be used to reproduce our results, additional adjustments may be possible to further improve our results. The code associated with this paper is available at `https://github.com/antn2023/ANTN`

**Implementation.** The transformer (Vaswani et al., 2017) used in this work is a decoder-only transformer implemented in (Luo et al., 2020), and the PixelCNN used is the gated PixelCNN (Van den Oord et al., 2016) implemented in (Chen et al., 2022) but without the channelwise mask.

**Hyperparameters.** For the transformer in quantum state learning, we use 2 layers, 32 hidden dimensions, and 16 attention heads, whereas for the PixelCNN in variational Monte Carlo, we use 7 layers with dilations 1, 2, 1, 4, 1, 2, and 1 for each layer and 48 hidden dimensions. For additional experiments with more layers, we always use dilation of 1 for the layer 8 and beyond.

**Initialization.** Throughout the experiment, we initialize the weight matrices of the last fully connected layer of the ARNN and ANTN to 0, and initialize the bias according to the following rule:

- The bias of ARNN is always randomly initialized.
- For the random Bell state learning, the bias of ANTN is randomly initialized

- For the rest of the experiment, the bias of ANTN is set to 0.

The rest of the parameters of ARNN and ANTN are randomly initialized according to PyTorch (Paszke et al., 2019)'s default initialization scheme.

In addition, for the random Bell state learning experiment, we do not initialize the ANTN with MPS (and hence the random bias), whereas for the shallow random circuit learning experiment, we initialize the ANTN with an MPS that comes from explicit truncation of the full wavefunction, and for the variational Monte Carlo experiment, we initialize the ANTN with DMRG optimized MPS.

**Optimization.** Through the experiment, we use the Adam optimizer with an initial learning rate of 0.01. For quantum state learning experiments, we train the ARNN and ANTN with full basis for 2000 iterations, where the learning rate is halved at iterations 600, 1200, and 1800. In case the full basis cannot be computed in one pass due to GPU memory constraint, we divide the full basis into several batches and accumulates the gradient in each iteration. For variational Monte Carlo experiments, we train the ARNN and ANTN stochastically until the energy converges. During the training, we halve the learning rate at iterations 100, 500, 1000, 1800, 2500, and 4000. In each experiment, we choose the maximum batch size that can be fed into the GPU memory, while using accumulation steps to keep the effective batch size around 10000 throughout the training.

**Transfer Learning.** We in addition take advantage of transfer learning. For $J_2 = 0.2$, 0.5, and 0.8, we begin from scratch with the $4 \times 4$ system. Then, an $N \times N$ system is initialized from an $(N - 2) \times (N - 2)$ system by keeping all but the weights in the last layer. Then, for other $J_2$, the neural network is initialized from one of $J_2 = 0.2$, 0.5, and 0.8 that is closest to the current $J_2$ of the same system size by keeping all but the weights in the last layer. We find this allows the optimization to converge faster than always starting from scratch.

**Computational Resources.** The models are mainly trained using NVIDIA V100 GPUs with 32 GB memory and Intel Xeon Gold 6248 CPUs, with some models trained using NVIDIA A100 GPUs with 80 GB memory.

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
