# OpenReview forum: "ANTN: Bridging Autoregressive Neural Networks and Tensor Networks for Quantum Many-Body Simulation"
_NeurIPS.cc/2023/Conference — NeurIPS 2023 poster_

### Official Review · Reviewer_wT6D · 2023-07-06

**Soundness:** 3 good
**Presentation:** 3 good
**Contribution:** 3 good
**Rating:** 7
**Confidence:** 4

**Summary:**

The paper approaches the problem of designing wave function ansatz for the ground state estimation of the Heisenberg model. This problem introduces several challenges including the rich parameterization of the wave function, sampling of the states from the parameterized wave function, and incorporating symmetries into the model. There are two main approaches to this problem: autoregressive neural networks and tensor networks.
The former allows for fast sampling and incorporating symmetries. The latter incorporates problem-specific inductive bias.

The authors propose an autoregressive model which incorporates tensor networks thus, hopefully, inheriting the best from both approaches. Shortly, the proposed model at every step of auto-regression yields tensors instead of scalars. This allows for a richer class of wave function representations while enjoying the properties of auto-regressive models such as exact sampling.

The authors study the proposed model both theoretically and empirically. The theoretical contribution of the paper is the proof of exact sampling and invariance properties. For the empirical study, the authors demonstrate the performance of the proposed ansatz for quantum state learning (fidelity maximization) and for the Heisenberg model (variational Monte Carlo).

**Strengths:**

I would say that the paper corresponds to all NeurIPS standards:
- The paper approaches an important problem with clear applications.
- The proposed method is novel and sound.
- The experimental study is extensive.
- It is well-written.

**Weaknesses:**

My only concern regarding the empirical study is the comparison in the number of parameters. Indeed, the main goal of the paper is to propose a richer family of models, where we would expect better approximation properties for the same number of parameters as in concurrent approaches. I believe that this can be better illustrated with experiments by reporting the number of parameters along with the energy values. Right now, the authors fix the shapes for the tensor network and then add a neural network "on top" of it. Therefore, it is not very clear how to compare the models. Maybe adding a plot (e.g., number of parameters vs energy) would improve the readability of the experimental section.

Other comments:
- line 127. Typo (articles).
- line 144. Typo (the last index of $\alpha$).
- line 153. Typo (many MPS represent)
- line 160. Typo (up to qubit $j$). Also, the notation with the summation over $\alpha$ is not very clear here.
- line 271. Typo.

**Questions:**

I have no questions for the authors.

**Limitations:**

The limitations are adequately addressed.

---

> ### Author Rebuttal · Authors · 2023-08-10
>
> > **Summary:**
> > The paper approaches the problem of designing wave function ansatz for the ground state estimation of the Heisenberg model. This problem introduces several challenges including the rich parameterization of the wave function, sampling of the states from the parameterized wave function, and incorporating symmetries into the model. There are two main approaches to this problem: autoregressive neural networks and tensor networks. The former allows for fast sampling and incorporating symmetries. The latter incorporates problem-specific inductive bias.
>
> > The authors propose an autoregressive model which incorporates tensor networks thus, hopefully, inheriting the best from both approaches. Shortly, the proposed model at every step of auto-regression yields tensors instead of scalars. This allows for a richer class of wave function representations while enjoying the properties of auto-regressive models such as exact sampling.
>
> > The authors study the proposed model both theoretically and empirically. The theoretical contribution of the paper is the proof of exact sampling and invariance properties. For the empirical study, the authors demonstrate the performance of the proposed ansatz for quantum state learning (fidelity maximization) and for the Heisenberg model (variational Monte Carlo).
>
> > **Strengths:**
> > I would say that the paper corresponds to all NeurIPS standards:
>
> > The paper approaches an important problem with clear applications.
> > The proposed method is novel and sound.
> > The experimental study is extensive.
> > It is well-written.
>
> **The authors thank the reviewer for the appreciation of this work.**
>
> > **Weaknesses:**
> > My only concern regarding the empirical study is the comparison in the number of parameters. Indeed, the main goal of the paper is to propose a richer family of models, where we would expect better approximation properties for the same number of parameters as in concurrent approaches. I believe that this can be better illustrated with experiments by reporting the number of parameters along with the energy values. Right now, the authors fix the shapes for the tensor network and then add a neural network "on top" of it. Therefore, it is not very clear how to compare the models. Maybe adding a plot (e.g., number of parameters vs energy) would improve the readability of the experimental section.
>
> **Thanks for the comment. While it is hard to set the number of parameters of all the ansatz exactly the same, to address the question, we have conducted additional experiments with additional parameters in ARNN, and list the energies, number of parameters, and runtimes for the experiments conducted in this work in the tables and the figures in the global response. We note that our ANTN actually has a comparable number of parameters to ARNN and has much fewer number of parameters than bond 1024 DMRG, while at the same time achieving better performance than all of them. The results further confirm that our ANTN achieves efficient representation and optimization by integrating tensor network and ARNN.**
>
>
> > **Other comments:**
>
> >line 127. Typo (articles).
> >line 144. Typo (the last index of $\alpha$).
> >line 153. Typo (many MPS represent)
> >line 160. Typo (up to qubit $j$). Also, the notation with the summation over $\alpha$ is not very clear here.
> >line 271. Typo.
>
> **The authors thank the reviewer for pointing out the typos and we have fixed them in the updated version of the paper.**
>
>
> >**Questions:**
> >I have no questions for the authors.
>
> >**Limitations:**
> >The limitations are adequately addressed.

---

> > ### Comment · Reviewer_wT6D · 2023-08-12
> >
> > Thank you for the response! I went over the rebuttal and I would like to keep my score.

---

> > > ### Author Response · Authors · 2023-08-20
> > >
> > > Thanks again for the valuable feedback! We have performed additional experiments comparing ANTN with ARNN to gain more understanding on their expressivity and optimization based on the suggestions during the discussion.
> > >
> > > We have tested a series of ARNN with different layers (up to 20 layers) and hidden dimensions (up to 72 hidden dimensions) such that the largest ARNN tested has more parameters than ANTN. We found that all the ARNN energies are not as good at ANTN (which is only based on 7 layers and 48 hidden dimensions), despite that ARNNs have more parameters and took longer time to train using the same GPU (see new data below). Furthermore, we observe that an increase of parameters can help ARNN to improve energy in general, but this improvement hits a diminishing return potentially due to difficulties in optimization as the number of parameters increases. In addition, we further test the (approximate) sign rule and found that it could help ARNN to obtain better energies but still worse than ANTN without sign rule (see Table 1 in the original paper and new data below). This indicates that the ANTN has the advantage of inheriting the flexible sign structure from MPS, avoiding the manual bias of adding the sign rule. The new results have further provided nice support for our work, both from an expressivity perspective and from the fact that ANTN has a better physics inductive bias than ARNN for optimization. We have also added the new benchmark in the updated manuscript.
> > >
> > > We hope these additional benchmarks will further resolve the concerns, and would be very appreciative if the reviewer could consider increasing the evaluation.

---

> > > > ### Author Response · Authors · 2023-08-20
> > > >
> > > > | Algorithm          | Energy per site | Number of layers | Hidden dimension | Bond dimension | Number of parameters (TN) | Number of parameters (NN) | Number of parameters (Total) | Runtime                        |
> > > > |--------------------|-----------------|------------------|------------------|----------------|---------------------------|---------------------------|------------------------------|--------------------------------|
> > > > | PixelCNN           | -1.74098(29)    | 7                | 48               | -              | 0                         | $5.0\times 10^5$          | $5.0\times 10^5$             | 19 hrs (A100) or 43 hrs (V100) |
> > > > | PixelCNN           | -1.86922(16)    | 8                | 48               | -              | 0                         | $5.7\times 10^5$          | $5.7\times 10^5$             | 59 hrs (V100)                  |
> > > > | PixelCNN           | -1.90021(15)    | 9                | 48               | -              | 0                         | $6.3\times 10^5$          | $6.3\times 10^5$             | 21 hrs (A100)                  |
> > > > | PixelCNN           | -1.90440(14)    | 10               | 48               | -              | 0                         | $7.0\times 10^5$          | $7.0 \times 10^5$            | 23 hrs (A100)                  |
> > > > | PixelCNN           | -1.92826(11)    | 11               | 48               | -              | 0                         | $7.7 \times 10^5$         | $7.7 \times 10^5$            | 26 hrs (A100)                  |
> > > > | PixelCNN           | -1.92986(10)    | 12               | 48               | -              | 0                         | $8.4 \times 10^5$         | $8.4\times 10^5$             | 30 hrs (A100)                  |
> > > > | PixelCNN           | -1.92869(11)    | 15               | 48               | -              | 0                         | $1.0\times 10^6$          | $1.0 \times 10^6$            | 51 hrs (A100)                  |
> > > > | PixelCNN           | -1.91537(13)    | 20               | 48               | -              | 0                         | $1.4 \times 10^6$         | $1.4 \times 10^6$            | 66 hrs (A100)                  |
> > > > | PixelCNN           | -1.88536(19)    | 8                | 72               | -              | 0                         | $1.3 \times 10^6$         | $1.3 \times 10^6$            | 35 hrs (A100)                  |
> > > > | PixelCNN           | -1.89266(15)    | 9                | 72               | -              | 0                         | $1.4 \times 10^6$         | $1.4 \times 10^6$            | 40 hrs (A100)                  |
> > > > | PixelCNN           | -1.90718(15)    | 10               | 72               | -              | 0                         | $1.6 \times 10^6$         | $1.6 \times 10^6$            | 47 hrs (A100)                  |
> > > > | PixelCNN           | -1.92965(13)    | 11               | 72               | -              | 0                         | $1.7 \times 10^6$         | $1.7 \times 10^6$            | 52 hrs (A100)                  |
> > > > | PixelCNN           | -1.91186(14)    | 12               | 72               | -              | 0                         | $1.9 \times 10^6$         | $1.9 \times 10^6$            | 57 hrs (A100)                  |
> > > > | PixelCNN Sign Rule | -1.92915(12)    | 11               | 48               | -              | 0                         | $7.7 \times 10^5$         | $7.7 \times 10^5$            | 31 hrs (A100)                  |
> > > > | PixelCNN Sign Rule | -1.93112(11)    | 12               | 48               | -              | 0                         | $8.4 \times 10^5$         | $8.4 \times 10^5$            | 34 hrs (A100)                  |
> > > > | PixelCNN Sign Rule | -1.93233(9)     | 15               | 48               | -              | 0                         | $1.0\times 10^6$          | $1.0\times 10^6$             | 47 hrs (A100)                  |
> > > > | PixelCNN Sign Rule | -1.93058(10)    | 20               | 48               | -              | 0                         | $1.4 \times 10^6$         | $1.4 \times 10^6$            | 64 hrs (A100)                  |
> > > > | PixelCNN Sign Rule | -1.93211(10)    | 11               | 72               | -              | 0                         | $1.7 \times 10^6$         | $1.7 \times 10^6$            | 54 hrs (A100)                  |
> > > > | PixelCNN Sign Rule | -1.92812(12)    | 12               | 72               | -              | 0                         | $1.9 \times 10^6$         | $1.9 \times 10^6$            | 59 hrs (A100)                  |

---

> > > > > ### Author Response · Authors · 2023-08-20
> > > > >
> > > > > | Algorithm          | Energy per site | Number of layers | Hidden dimension | Bond dimension | Number of parameters (TN) | Number of parameters (NN) | Number of parameters (Total) | Runtime                        |
> > > > > |--------------------|-----------------|------------------|------------------|----------------|---------------------------|---------------------------|------------------------------|--------------------------------|
> > > > > | DMRG               | -1.734326       | -                | -                | 8              | $1.2\times 10^4$          | 0                         | $1.2 \times 10^4$            | < 1 hr                         |
> > > > > | DMRG               | -1.866159       | -                | -                | 70             | $8.8\times 10^5$          | 0                         | $8.8 \times 10^5$            | < 1 hr                         |
> > > > > | DMRG               | -1.938770       | -                | -                | 1024           | $1.7\times 10^8$          | 0                         | $1.7 \times 10^8$            | 242 hrs                        |
> > > > > | Elementwise        | -1.94001(6)     | 7                | 48               | 8              | $1.2 \times 10^6$         | $5.0\times 10^5$          | $1.7 \times 10^6$            | 46 hrs (A100) or 89 hrs (V100) |
> > > > > | Blockwise          | -1.93842(42)    | 7                | 48               | 70             | $8.9 \times 10^5$         | $5.0 \times 10^5$         | $1.4 \times 10^6$            | 136 hrs (V100)                 |

---

### Official Review · Reviewer_dUKK · 2023-07-07

**Soundness:** 3 good
**Presentation:** 3 good
**Contribution:** 3 good
**Rating:** 5
**Confidence:** 3

**Summary:**

The paper proposes generic Autoregressive Neural TensorNet (ANTN), for quantum many-body simulation. In order to achieve high expressivity, sign structure preserving, physics inductive bias, accurate sampling, and symmetry preservation, ANTN combines tensor networks and autoregressive neural networks. The author demonstrates the improved performance of ANTN on quantum state learning and learning ground state energy.

**Strengths:**

1. The proposed architecture absorbs the advantages of two state-of-the-art methods i.e., tensor networks and autoregressive neural networks. The idea is straightforward but novel and interesting.
2. This paper well illustrates its differences and advantages over tensor network based and neural network quantum state based methods for quantum many body simulation.
3. It provides a comprehensive theoretical analysis and the experimental results show that the proposed method can still maintain good expressive ability with fewer bond dimensions.

**Weaknesses:**

1. The author solely compares and reports the bond dimensions taken up by the ANTN and MPS/DMRG approaches in Fig 1, Tab 1 and 2. Given that the first half of the ANTN is a sophisticated autoregressive neural network, reviewer concerns about that the author may evade a fact that whether the order of magnitude of the parameters of the autoregressive neural network could surpass the bond dimension of the following tensor network. As a consequence, the existence of the NN parameters will cancel out the efficiency brought on by the reduction in bond dimension.
2. Although the combination of autoregressive neural network and tensor network to build a new neural network quantum state method that absorbs the advantages of both is a delightful idea, reviewers are worried that such method will also inherit the disadvantages of both. For example, tensor networks are likely to be very tricky to characterize a highly entangled quantum state [1]. At the same time, traditional tensor networks may also perform poorly in learning the ground state energy of a long-range Hamiltonian [2,3]. These problems are also an important reason why some researches abandon the tensor network and embrace the neural network-based quantum states. The reviewer is pleased if the authors could clarify these points, and it would be preferable if they could provide some explanations and comparisons in the experimental or theoretical part.

[1] Deng D L, Li X, Sarma S D. Quantum entanglement in neural network states[J]. Physical Review X, 2017, 7(2): 021021.

[2] Glasser I, Pancotti N, August M, et al. Neural-network quantum states, string-bond states, and chiral topological states[J]. Physical Review X, 2018, 8(1): 011006.

[3] Xiao T, Huang J, Li H, et al. Intelligent certification for quantum simulators via machine learning[J]. npj Quantum Information, 2022, 8(1): 138.

**Questions:**

Please see the weaknesses.

**Limitations:**

The author does describe the limitations.

---

> ### Author Rebuttal · Authors · 2023-08-10
>
> >**Summary:**
> >The paper proposes generic Autoregressive Neural TensorNet (ANTN), for quantum many-body simulation. In order to achieve high expressivity, sign structure preserving, physics inductive bias, accurate sampling, and symmetry preservation, ANTN combines tensor networks and autoregressive neural networks. The author demonstrates the improved performance of ANTN on quantum state learning and learning ground state energy.
>
> >**Strengths:**
> >The proposed architecture absorbs the advantages of two state-of-the-art methods i.e., tensor networks and autoregressive neural networks. The idea is straightforward but novel and interesting.
> >This paper well illustrates its differences and advantages over tensor network based and neural network quantum state based methods for quantum many body simulation.
> >It provides a comprehensive theoretical analysis and the experimental results show that the proposed method can still maintain good expressive ability with fewer bond dimensions.
>
> **The authors thank the reviewer for the appreciation of this work.**
>
> >**Weaknesses:**
> >The author solely compares and reports the bond dimensions taken up by the ANTN and MPS/DMRG approaches in Fig 1, Tab 1 and 2. Given that the first half of the ANTN is a sophisticated autoregressive neural network, reviewer concerns about that the author may evade a fact that whether the order of magnitude of the parameters of the autoregressive neural network could surpass the bond dimension of the following tensor network. As a consequence, the existence of the NN parameters will cancel out the efficiency brought on by the reduction in bond dimension.
>
> **The authors thank the reviewer for bringing up the question regarding the number of parameters for the MPS, ARNN and ANTN. The authors would like to clarify that for the experiments that we conducted, MPS with the bond dimension of $1024$ actually has the most parameters ($10^8$), given that the number of parameters for MPS scales as $N\chi^2$ with $N$ the system size and $\chi$ the bond dimension. The ARNN has fewer parameters ($5.0\times 10^5$) compared to the ANTN ($10^6$), because we fixed the shape of the underlying ARNN when conducting the experiment. In the global response, we list out in detail the number of parameters and runtimes for the experiments. We find that our ANTN outperforms both ARNN and MPS even though ANTN has a comparable number of parameters as ARNN and much fewer parameters than MPS.**
>
> >Although the combination of autoregressive neural network and tensor network to build a new neural network quantum state method that absorbs the advantages of both is a delightful idea, reviewers are worried that such method will also inherit the disadvantages of both. For example, tensor networks are likely to be very tricky to characterize a highly entangled quantum state [1]. At the same time, traditional tensor networks may also perform poorly in learning the ground state energy of a long-range Hamiltonian [2,3]. These problems are also an important reason why some researches abandon the tensor network and embrace the neural network-based quantum states. The reviewer is pleased if the authors could clarify these points, and it would be preferable if they could provide some explanations and comparisons in the experimental or theoretical part.
>
> **Thanks for the question. The authors understand the reviewer's concern about combining ARNN and MPS. However, as shown in Theorem 5.2, the ANTN has generalized expressivity over both ARNN and MPS. In fact, both ARNN and MPS are special cases of ANTN. Therefore, ANTN does not inherit the disadvantages of both ARNN and MPS.  Moreover, we updated Theorem 5.2 in the global response, where we explicitly proved that neither ARNN nor MPS can efficiently represent ANTN. In addition, ANTN inherits the flexible sign structure from tensor network, which plays an important role for optimization. As we show in the new experiments in the one-page response, our ANTN achieves the best performance in those cases compared to RBM and ARNN with both sign rule and non-sign rule as well as MPS. Our new theoretical and experimental results strongly support that ANTN takes advantages of both MPS and ARNN. Regarding the specific example that the reviewer pointed out. ANTN is able to capture the entanglement very well. As we demonstrated in Figure 2(a), the ANTN achieves the same fidelity as the ARNN for random Bell states (maximally entangled). In an ANTN, the MPS and ARNN divide the work of representing a quantum state, with the MPS specializing in representing the quasi-local sign structure (Figure 2 (b)), and the ARNN focusing on representing the entanglement. Therefore, the ANTN actually inherits the advantages of both ARNN and MPS, which is exactly the reason that ANTN surpasses both MPS and ARNN in our experiments. We hope this explanation resolves the reviewer's concern.**
>
> >**Questions:**
> >Please see the weaknesses.
>
> >**Limitations:**
> >The author does describe the limitations.

---

> > ### Author Response · Authors · 2023-08-20
> >
> > Thanks again for the valuable feedback! We have performed additional experiments comparing ANTN with ARNN to gain more understanding on their expressivity and optimization based on the suggestions during the discussion.
> >
> > We have tested a series of ARNN with different layers (up to 20 layers) and hidden dimensions (up to 72 hidden dimensions) such that the largest ARNN tested has more parameters than ANTN. We found that all the ARNN energies are not as good at ANTN (which is only based on 7 layers and 48 hidden dimensions), despite that ARNNs have more parameters and took longer time to train using the same GPU (see new data below). Furthermore, we observe that an increase of parameters can help ARNN to improve energy in general, but this improvement hits a diminishing return potentially due to difficulties in optimization as the number of parameters increases. In addition, we further test the (approximate) sign rule and found that it could help ARNN to obtain better energies but still worse than ANTN without sign rule (see Table 1 in the original paper and new data below). This indicates that the ANTN has the advantage of inheriting the flexible sign structure from MPS, avoiding the manual bias of adding the sign rule. The new results have further provided nice support for our work, both from an expressivity perspective and from the fact that ANTN has a better physics inductive bias than ARNN for optimization. We have also added the new benchmark in the updated manuscript.
> >
> > We hope these additional benchmarks will further resolve the concerns, and would be very appreciative if the reviewer could consider increasing the evaluation.

---

> > > ### Author Response · Authors · 2023-08-20
> > >
> > > | Algorithm          | Energy per site | Number of layers | Hidden dimension | Bond dimension | Number of parameters (TN) | Number of parameters (NN) | Number of parameters (Total) | Runtime                        |
> > > |--------------------|-----------------|------------------|------------------|----------------|---------------------------|---------------------------|------------------------------|--------------------------------|
> > > | PixelCNN           | -1.74098(29)    | 7                | 48               | -              | 0                         | $5.0\times 10^5$          | $5.0\times 10^5$             | 19 hrs (A100) or 43 hrs (V100) |
> > > | PixelCNN           | -1.86922(16)    | 8                | 48               | -              | 0                         | $5.7\times 10^5$          | $5.7\times 10^5$             | 59 hrs (V100)                  |
> > > | PixelCNN           | -1.90021(15)    | 9                | 48               | -              | 0                         | $6.3\times 10^5$          | $6.3\times 10^5$             | 21 hrs (A100)                  |
> > > | PixelCNN           | -1.90440(14)    | 10               | 48               | -              | 0                         | $7.0\times 10^5$          | $7.0 \times 10^5$            | 23 hrs (A100)                  |
> > > | PixelCNN           | -1.92826(11)    | 11               | 48               | -              | 0                         | $7.7 \times 10^5$         | $7.7 \times 10^5$            | 26 hrs (A100)                  |
> > > | PixelCNN           | -1.92986(10)    | 12               | 48               | -              | 0                         | $8.4 \times 10^5$         | $8.4\times 10^5$             | 30 hrs (A100)                  |
> > > | PixelCNN           | -1.92869(11)    | 15               | 48               | -              | 0                         | $1.0\times 10^6$          | $1.0 \times 10^6$            | 51 hrs (A100)                  |
> > > | PixelCNN           | -1.91537(13)    | 20               | 48               | -              | 0                         | $1.4 \times 10^6$         | $1.4 \times 10^6$            | 66 hrs (A100)                  |
> > > | PixelCNN           | -1.88536(19)    | 8                | 72               | -              | 0                         | $1.3 \times 10^6$         | $1.3 \times 10^6$            | 35 hrs (A100)                  |
> > > | PixelCNN           | -1.89266(15)    | 9                | 72               | -              | 0                         | $1.4 \times 10^6$         | $1.4 \times 10^6$            | 40 hrs (A100)                  |
> > > | PixelCNN           | -1.90718(15)    | 10               | 72               | -              | 0                         | $1.6 \times 10^6$         | $1.6 \times 10^6$            | 47 hrs (A100)                  |
> > > | PixelCNN           | -1.92965(13)    | 11               | 72               | -              | 0                         | $1.7 \times 10^6$         | $1.7 \times 10^6$            | 52 hrs (A100)                  |
> > > | PixelCNN           | -1.91186(14)    | 12               | 72               | -              | 0                         | $1.9 \times 10^6$         | $1.9 \times 10^6$            | 57 hrs (A100)                  |
> > > | PixelCNN Sign Rule | -1.92915(12)    | 11               | 48               | -              | 0                         | $7.7 \times 10^5$         | $7.7 \times 10^5$            | 31 hrs (A100)                  |
> > > | PixelCNN Sign Rule | -1.93112(11)    | 12               | 48               | -              | 0                         | $8.4 \times 10^5$         | $8.4 \times 10^5$            | 34 hrs (A100)                  |
> > > | PixelCNN Sign Rule | -1.93233(9)     | 15               | 48               | -              | 0                         | $1.0\times 10^6$          | $1.0\times 10^6$             | 47 hrs (A100)                  |
> > > | PixelCNN Sign Rule | -1.93058(10)    | 20               | 48               | -              | 0                         | $1.4 \times 10^6$         | $1.4 \times 10^6$            | 64 hrs (A100)                  |
> > > | PixelCNN Sign Rule | -1.93211(10)    | 11               | 72               | -              | 0                         | $1.7 \times 10^6$         | $1.7 \times 10^6$            | 54 hrs (A100)                  |
> > > | PixelCNN Sign Rule | -1.92812(12)    | 12               | 72               | -              | 0                         | $1.9 \times 10^6$         | $1.9 \times 10^6$            | 59 hrs (A100)                  |

---

> > > > ### Author Response · Authors · 2023-08-20
> > > >
> > > > | Algorithm          | Energy per site | Number of layers | Hidden dimension | Bond dimension | Number of parameters (TN) | Number of parameters (NN) | Number of parameters (Total) | Runtime                        |
> > > > |--------------------|-----------------|------------------|------------------|----------------|---------------------------|---------------------------|------------------------------|--------------------------------|
> > > > | DMRG               | -1.734326       | -                | -                | 8              | $1.2\times 10^4$          | 0                         | $1.2 \times 10^4$            | < 1 hr                         |
> > > > | DMRG               | -1.866159       | -                | -                | 70             | $8.8\times 10^5$          | 0                         | $8.8 \times 10^5$            | < 1 hr                         |
> > > > | DMRG               | -1.938770       | -                | -                | 1024           | $1.7\times 10^8$          | 0                         | $1.7 \times 10^8$            | 242 hrs                        |
> > > > | Elementwise        | -1.94001(6)     | 7                | 48               | 8              | $1.2 \times 10^6$         | $5.0\times 10^5$          | $1.7 \times 10^6$            | 46 hrs (A100) or 89 hrs (V100) |
> > > > | Blockwise          | -1.93842(42)    | 7                | 48               | 70             | $8.9 \times 10^5$         | $5.0 \times 10^5$         | $1.4 \times 10^6$            | 136 hrs (V100)                 |

---

### Official Review · Reviewer_Vtcz · 2023-07-10

**Soundness:** 3 good
**Presentation:** 4 excellent
**Contribution:** 3 good
**Rating:** 7
**Confidence:** 4

**Summary:**

In this work, the authors introduce ANTN (Autoregressive Neural Tensor Net) as a novel architecture that combines features from both ARNN and Tensor Networks for simulating challenging quantum many-body systems. The main objective of this research is to utilize an ARNN-type network to generate a conditional wavefunction tensor rather than a direct conditional wavefunction, as commonly done in standard ARNNs. This latter can be interpreted as a probability density. Figure 1 provides a clear visualization of the idea.

By integrating ARNN and TN within this framework, the authors aim to leverage the advantages of both methods, thereby achieving enhanced expressivity and a stronger adherence to physics inductive bias. To demonstrate the superiority of their approach, the authors present several theorems highlighting its advantages. While detailed proofs are available in the appendix, the main text often provides concise explanations, which significantly support the flow of the manuscript and complement the discussed results.

The method's performance is initially evaluated by benchmarking its ability to learn 2-qubit Bell random states and real-valued shallow random circuits. Finally, the authors compare their ANTN approach to state-of-the-art methods in solving the ground state of a 2D $J_1-J_2$ Heisenberg model with open boundary conditions. This particular problem is known to be challenging due to its phase diagram exhibiting at least three distinct phases as $J_1$ and $J_2$ are varied. The results presented in Table 1 and Table 2 demonstrate that the ANTN method consistently outperforms existing methods across different system sizes and parameter choices for $J_2$.

**Strengths:**

- The paper is scientifically sound and well-written, with a comprehensive and meticulous literature review that covers all relevant works in the field.
- The paper's structure is highly effective, resulting in a smooth and enjoyable reading experience.
- Section 3 stands out as it provides a thorough overview of the essential components required to comprehend the more theoretical discussion starting from Section 4. This inclusion enhances the accessibility of the paper, making it approachable even for readers who are not thoroughly familiar with the field, such as myself.
- The concept of combining the inherent flexibility of ARNN with the strengths of TN within a single framework is undeniably compelling.

**Weaknesses:**

- The manuscript frequently mentions the concept of "expressivity," but it is never explicitly defined or explained. It would greatly benefit the readers to dedicate a paragraph to clarify its meaning and highlight its significance, thereby implicitly assessing the advantages of the proposed approach. Specifically, Theorem 5.2 is not entirely clear to me. It would be helpful to have a clear definition of expressivity to shed light on this point.
- The paper repeatedly refers to the "sign structure" problem, but this issue is never explicitly defined or explained. While referencing relevant literature is helpful, the paper should strive to be self-contained when discussing this critical aspect. Given its importance in evaluating the efficacy of the proposed method in the experimental section, a paragraph discussing this concept would be valuable for readers less familiar with it.
- Although ANTN combines properties from both ARNN and TN, being a parametrized neural network, it is expected that ANTN also requires training. However, the main text does not discuss this in great detail. A comprehensive discussion on training ANTN in the main text is essential to provide a complete understanding of the methodology.
- The manuscript lacks experiments that benchmark the runtime and memory complexity. It would be beneficial to compare ANTN with state-of-the-art TN methods in terms of these metrics. Such comparisons would enhance the overall evaluation of ANTN and its performance.

**Questions:**

- How is the ANTN model trained? While intuitions are provided in Sections 3.2 and 3.3 (with additional details in the appendix), a more specific discussion would enhance understanding of the proposed framework. Additionally, it would be beneficial to explore the limitations of training ANTN. Is the training process significantly more challenging compared to plain ARNN? Can you provide an estimate of the computational time required to train a network for a specific downstream task?
- The authors frequently mention that ARNNs lack the physics prior of the system under study. However, it raises the question of the extent to which this holds true. As correctly stated by the authors, inductive physics biases, such as symmetries, can actually be incorporated into pure ARNN networks to incorporate the desired physics prior. Could the authors provide some comments on this?
- I wonder whether it would be interesting to replicate the analysis in Table 1 and Table 2 by fixing $J_2$ and modifying $J_1$ instead. Would substantial differences in the results be expected? This question arises because, as I understand it, $J_1$ modulates the interaction between nearest neighbors, potentially leading to collective behavior that may be more challenging to capture.
- How do runtime and memory requirements scale in the experiments presented in Section 6.2? The insightful complexity analysis discussed in Section 5 suggests that it would be valuable to empirically visualize these factors through numerical experiments.
- Is there a specific reason why DGRM results are not shown in Table 1, while RBM results are missing in Table 2?
- In Figure 3, it appears that the DGRM method outperforms ANTN for medium-sized systems and increasing values of $J_2$. Since the bond dimension is fixed for both methods, do you have any intuition regarding how this plot would change as the bond dimension varies? Intuitively, one might expect that having a larger bond state for DGRM would improve its performance, particularly for larger system sizes. It would be intriguing to perform a detailed scaling analysis as a function of the bond dimension, acknowledging the potential increase in complexity and the enhanced capability of TN methods to capture more features, potentially leading to improved results.
- In the broader impact section, the authors suggest that the ANTN approach may be applicable to other standard machine learning tasks. However, in such tasks, like image generation, for instance, there might not be strong physics prior to leverage using the TN representation. It would be interesting to hear the authors' perspectives on the benefits of applying their framework to standard machine learning benchmarks and how it could be advantageous in those scenarios.
- Can this idea of combining TN with a generative type of neural networks be extended to other generative models?

**Limitations:**

In addition to the concerns mentioned in the weaknesses and questions sections, which implicitly highlight some limitations of the current manuscript, I believe that the primary limitation of this work may lie in its scalability and the training difficulties of ANTN compared to MPS. Although I am not an MPS expert, I am aware that sampling (and training) for an ARNN can be slow and computationally expensive, particularly when dealing with larger systems. While runtime may also pose a challenge for MPS, as increasing the bond dimension can become a computational bottleneck, I am uncertain whether this would be a minor issue compared to training, and sampling from, an ANTN when considering extremely large systems.

A few minor comments:
- In the introduction, it would be good to already have the relevant references being cited when a field/topic is firstly mentioned and introduced.
- The Acronyms MERA and PEPS are never introduced in their extensive form. Would be good to do so when they are mentioned for the first time in line 72 of page 2.
- Line 271 page 7:  previous -> previously
- Line 287 page 7:  comes -> come
- Line 690 page 19: staring -> starting

---

> ### Author Rebuttal · Authors · 2023-08-10
>
>
> **Thanks for the appreciation of our work. We apologize that due to the character limit, we have to simply the response. We will provide additional explanations in the follow-up discussion.**
>
> >**Weaknesses:**
>
> >The manuscript frequently mentions the concept of "expressivity," .....
>
> **Thanks for the comment. Indeed, there could be different definitions of "expressivity" based on different perspectives. Two typical important ideas of expressivity are related to entanglement and sign structure, which could be both incorporated into ANTN. In particular, the degree of entanglement is the bottleneck for MPS since it is designed to represent low entanglement quantum wave functions. The sign structure (or more generally the phase structure) stems from the fact that quantum wave function can have complex-number output, which requires more design of the neural network architecture. In this manuscript, we choose to define "expressivity" in a general way as a neural network's ability to represent (generic) quantum wave functions (potentially limiting the number of parameters). We also updated the proof of Theorem 5.2 in the global response. We have included the above discussion on expressivity in the updated paper.**
>
> >The paper repeatedly refers to the "sign structure" problem.....
>
> **Thanks for the suggestion. The sign structure (or more generally the phase structure) comes from the fact that quantum wave functions are L2-normalized functions. In other words, a quantum wave function can be viewed as a probability distribution with complex-valued phases ($\psi(x)=\sqrt{p(x)}\exp{i\theta(x)}$). For real-valued Hamiltonian, such phases are reduced to signs of $+1$ and $-1$ ($\theta(x)=0$ or $\pi$) (i.e. the sign structure). To capture such a structure, the neural network requires additional design to go beyond a probability representation. When the sign structure is known, the quantum wave function can be reduced to the probability distribution and can be solved more easily. However, such a structure is only analytically known for a limited number of models, and in the worst case it can be very random binary number distribution of $+1$ and $-1$ that are hard to represent. In this work, ANTN can learn this sign structure more efficiently due to the inductive bias from the MPS. We have included more discussion on the "sign structure" problem in the updated paper.**
>
> >Although ANTN combines properties......
>
> **The ANTN is trained using standard gradient-based optimizers (Adam to be specific) with the gradient derived in Appendix A.2. We have added this in the updated paper to be clearer to the readers.**
>
> >The manuscript lacks experiments......
>
> **Thanks for the comment. Yes, it would be very beneficial to provide benchmarks regarding runtime and memory complexity. A theoretical analysis is provided in Section 5.1. And additional experimental runtime results are included in the global response. In practice, all of the experiments expect DMRG in this work can be efficiently run on NVIDIA V100 GPU with 32 GB memory using at least 2000 samples in parallel. The DMRG algorithm with a bond dimension of 1024 requires a substantial amount of memory (with $10^8$ parameters) and the optimization is hard to be efficiently parallelized. Therefore it is run on CPU (which is standard for the DMRG algorithm.) We have included these discussions in the updated paper.**
>
> >**Questions:**
>
> >How is the ANTN model trained...
>
> **In Sections 3.2 and 3.3, we provide the training loss functions. For quantum state learning, the gradient is simply the gradient of the loss function. For the variation Monte Carlo algorithm, the gradient is similar to the policy gradient of the setup in reinforcement learning, which we derive in Appendix A.2. Then, Adam optimizer was used to train the neural network, with details written in Appendix C. We hope this answers the reviewer's question.**
>
> >The authors frequently mention that ARNNs lack...
>
> **As discussed previously, there are different perspectives of expressivity and inductive bias of a quantum wave function. While symmetry could be Incorporated into ARNN, there are other important inductive bias such as low entanglement and sign structure, which may not be easy to capture by ARNN. Indeed, it is known that MPS is better at capturing low entanglement (such as low dimension system ground state) and sign structure. Hence, by bridging MPS and ARNN together into ANTN, ANTN is equipped with more inductive bias, not only multiply symmetries from ARNN, but also sign structure and various entanglement degree (both low entanglement from MPS and high entanglement from ARNN).**
>
> >I wonder whether ...
>
> **Thanks for the comment. Since finding the ground state corresponds to finding the smallest eigenvalue of the Hamiltonian matrix, multiplying the matrix by a constant does not change the problem (it only rescales the eigenvalue). Hence, the only physically relevant quantity is the ratio $J_2/J_1$.**
>
> >How do runtime......
>
> **Already answered above**
>
> >Is there a specific reason why DGRM ..
>
> >In Figure 3, it appears that the DGRM..
>
> >In the broader impact section..
>
> **We will answer these questions in the follow-up discussion due to the character limit.**
>
> >**Limitations:**
>
> >In addition to...
>
> **The authors understand the concerns that the reviewer raises. However, ARNN (and ANTN) are not slower compared to MPS, especially with the ability of exact sampling. In fact, because ARNN and ANTN can be trained more efficiently on GPU compared to MPS (with DMRG algorithm) due to the lower memory requirement and easier parallelization, ARNN and ANTN can be faster in practice with runtime details included in the global response.**
>
> >**A few minor comments:**
>
> **Thanks for the suggestions and corrections, and we have updated the manuscript accordingly.**

---

> > ### Author Response · Authors · 2023-08-10
> > **Additional Responses**
> >
> > **We apologize for not being able to include all the responses above due to the character limit. Below, we will answer the remaining questions below.**
> >
> > >Is there a specific reason why DGRM results are not shown in Table 1, while RBM results are missing in Table 2?
> >
> > **Thanks for the question. Table 1 focuses on demonstrating the effect of the Marshall sign rule on different neural network architectures. Because DMRG algorithm is not affected by the sign rule by design, presenting the results is not necessary. As RBM is shown to be worse than PixelCNN in Table 1, it becomes unnecessary to further include the results on larger systems since it may dilute the main message of the table. We note that we indeed checked the RBM results for $10\times10$ systems and confirmed that the energies are not as good as either PixelCNN or ANTN.**
> >
> > >In Figure 3, it appears that the DGRM method outperforms ANTN for medium-sized systems and increasing values of $J_2$. Since the bond dimension is fixed for both methods, do you have any intuition regarding how this plot would change as the bond dimension varies? Intuitively, one might expect that having a larger bond state for DGRM would improve its performance, particularly for larger system sizes. It would be intriguing to perform a detailed scaling analysis as a function of the bond dimension, acknowledging the potential increase in complexity and the enhanced capability of TN methods to capture more features, potentially leading to improved results.
> >
> > **For large $J_2$, the system goes into a stripped phase as shown in (Nomura & Imada, 2021), which can decouple certain rows (columns) and therefore lowers the entanglement, where DMRG algorithm can be more advantageous. However, it is known that for 2D ground state in $L\times L$ system that satisfies area law, its entanglement grows with L and it requires MPS bond dimension to grow exponentially with L. The increase in entanglement for large systems would still make the DMRG algorithm more difficult in general.**
> >
> > >In the broader impact section, the authors suggest that the ANTN approach may be applicable to other standard machine learning tasks. However, in such tasks, like image generation, for instance, there might not be strong physics prior to leverage using the TN representation. It would be interesting to hear the authors' perspectives on the benefits of applying their framework to standard machine learning benchmarks and how it could be advantageous in those scenarios.
> >
> > >Can this idea of combining TN with a generative type of neural networks be extended to other generative models?
> >
> > **Thanks for the questions. TN algorithms have been investigated before in tasks such as image generation, and ANTN should be able to further enhance the performance due to more powerful representation. It will be an interesting direction to study how the ANTN performs in such tasks compared to standard NNs. It would also be meaningful to investigate how to combine TN with other types of neural networks, such as flow or diffusion models.**

---

> > > ### Comment · Reviewer_Vtcz · 2023-08-11
> > > **Acknowledgments for author's response**
> > >
> > > I sincerely appreciate the authors for taking the time to answer all the comments and questions I raised carefully.
> > >
> > > Looking at the one-page PDF document, the main conclusion that I draw is that the most significant advantage of the proposed ANTN approach is to reach (DMRG) performance level and beyond, substantially reducing the computational cost. Do the authors agree with this statement or do they think there might be more important benefits here?
> > >
> > > Also, in Figure R1, what do the authors mean by "Energy per site"? I am not sure what to gather from the figure.
> > >
> > > About my previous comments:
> > >
> > > **On the expressivity**
> > > Thanks for the clarification. Adding this discussion in the paper would clarify this to future readers and improve the paper.
> > >
> > > **On the additional experiments**
> > > That indeed adds a lot of information. Thank you. Have you tried to optimize the number of parameters needed for the MPS for ANTN such that it is minimized without affecting the overall learning capability? Looking at the table shown in the one-page PDF suggests to me there might be a sweet-spot worth finding. Based on their experience with the experiments, do the authors agree, or do they think this might not change the performance that much?
> > >
> > > **On the training**
> > > It is not yet fully clear to me how the training of the entire process is performed. I might have to read the relevant sections once again. However, I appreciate the effort of the authors in clarifying this.
> > >
> > > The regime for larger $J_2$ appears to be more challenging for MPS for the reasons given by the authors. I am curious if the authors expect that adding more layers and improving on the NN structure would compensate for this imperfection of capturing all the relevant physical prior from the TN part of the algorithm.
> > >
> > > Other than the comments above, which are rather pure curiosities rather than concerns, the authors have proven their work to be solid by satisfactorily responding to all my comments. Therefore, I now feel more confident than before this is good work and thus deserves acceptance. I'd therefore raise my confidence. However, I'd keep the score to 7 (Accept) for now. I'll use the remaining days to re-read the manuscript and decide on whether to raise the score as well.

---

> > > > ### Author Response · Authors · 2023-08-12
> > > > **Thanks for the feedback!**
> > > >
> > > > >I sincerely appreciate the authors for taking the time to answer all the comments and questions I raised carefully.
> > > >
> > > > **Thanks a lot for the feedback! We will answer the new questions the reviewer posts below.**
> > > >
> > > > >Looking at the one-page PDF document, the main conclusion that I draw is that the most significant advantage of the proposed ANTN approach is to reach (DMRG) performance level and beyond......
> > > >
> > > > **Yes, this is one of the most significant practical advantages of the ANTN as demonstrated, since DMRG is one of the state-of-the-arts methods and it is crucial to develop approach beyond given its limitation for higher dimensional systems. In addition, there are the other nice features of ANTN, such as being more expressive in general and inheriting varous properties from ARNN and MPS.**
> > > >
> > > > >Also, in Figure R1, what do the authors mean by "Energy per site"?......
> > > >
> > > > **Thanks for the question! We use same metric of energy per site throughout the paper in other tables. The energy per site is just the calculated ground state energy divided by the system size. This makes an easier comparison for different system sizes. We have now clarified this point in the updated manuscript. In Figure R1, we just want to visualize how the energy compares across different architectures with different numbers of parameters and runtimes.**
> > > >
> > > > >About my previous comments:
> > > >
> > > > >On the expressivity Thanks for the clarification......
> > > >
> > > > **Thanks! We have now added the discussion to the updated manuscript.**
> > > >
> > > > >On the additional experiments That indeed adds a lot of information. Thank you. Have you tried to optimize the number of parameters needed for the MPS for ANTN such that it is minimized without affecting the overall learning capability? Looking at the table......
> > > >
> > > > **Thanks for the questions. We also agree that there should be a sweep-spot. Actually the ARNN is a special case of ANTN as a bond=1 ANTN. Since our bond=8 ANTN is better than ARNN, it indicates that the increase of MPS bond dimension is definitely helpful. However, if MPS has too much parameters with very large bond dimension, it will increase the memory and optimization cost a lot, which may not have too much gain in practice. A detailed study on the effect of MPS parameters will be an intersting direction for the future work.**
> > > >
> > > >
> > > > >On the training It is not yet fully clear to me how the training of the entire process is performed......
> > > >
> > > > **Thanks for the question. For MPS, the training is done via the standard DMRG algorithm on CPU. For ARNN, we use the transfer learning technique with first pretraining the NN on 4x4 system using Adam optimizer with the gradient mentioned in Sec. 3.3 and A.2; then we keep all the weights of the NN except the last layer and move on to 6x6 system. The same procedure is used to further increase the system size from 6x6 to larger systems. For ANTN, we use the same transfer learning procedure, except that for each system size, the TN part of ANTN is always initialized with the DMRG result at that system size. We hope that this further clarifies the training process and we have now included the explanation in the updated manuscript. We kindly ask if the reviewer has any further questions regarding the training process and would be glad to make further clarification.**
> > > >
> > > > >The regime for larger $J_2$ appears to be more challenging for MPS for the reasons given by the authors......
> > > >
> > > > **We apologize for not making this point very clear. In our response, we were trying to say that larger $J_2$ is easier for MPS since the ground state at these points may have a relatively lower entanglement (even though it can still be challenging for MPS as system size grows). This makes DMRG better than the current ANTN at these points. We do expect that further increasing the parameters of ANTN could help capture all physics and make ANTN better than DMRG as it will increase the representation power. In addition, according to the trend of energy vs. system size, going to larger system sizes even with the same setup has already made ANTN better than DMRG in almost all cases (which is consistent with theoretical argument that DMRG fails as the system size increases and the entanglement increases).**
> > > >
> > > > >Other than the comments above, which are rather pure curiosities rather than concerns, the authors have proven their work to be solid by satisfactorily responding to all my comments. Therefore, I now feel more confident than before this is good work and thus deserves acceptance. I'd therefore raise my confidence. However, I'd keep the score to 7 (Accept)
> > > >
> > > > >for now. I'll use the remaining days to re-read the manuscript and decide on whether to raise the score as well.
> > > >
> > > > **Thanks a lot for raising the confidence level. We hope that we have addressed the questions above and would be very appreciative if the reviewer could consider raising the score later. It would also be our pleasure to answer any additional questions that the reviewer would like to discuss.**

---

> > > > > ### Author Response · Authors · 2023-08-20
> > > > >
> > > > > Thanks again for the valuable feedback! We have performed additional experiments comparing ANTN with ARNN to gain more understanding on their expressivity and optimization based on the suggestions during the discussion.
> > > > >
> > > > > We have tested a series of ARNN with different layers (up to 20 layers) and hidden dimensions (up to 72 hidden dimensions) such that the largest ARNN tested has more parameters than ANTN. We found that all the ARNN energies are not as good at ANTN (which is only based on 7 layers and 48 hidden dimensions), despite that ARNNs have more parameters and took longer time to train using the same GPU (see new data below). Furthermore, we observe that an increase of parameters can help ARNN to improve energy in general, but this improvement hits a diminishing return potentially due to difficulties in optimization as the number of parameters increases. In addition, we further test the (approximate) sign rule and found that it could help ARNN to obtain better energies but still worse than ANTN without sign rule (see Table 1 in the original paper and new data below). This indicates that the ANTN has the advantage of inheriting the flexible sign structure from MPS, avoiding the manual bias of adding the sign rule. The new results have further provided nice support for our work, both from an expressivity perspective and from the fact that ANTN has a better physics inductive bias than ARNN for optimization. We have also added the new benchmark in the updated manuscript.
> > > > >
> > > > > We hope these additional benchmarks will further resolve the concerns, and would be very appreciative if the reviewer could consider increasing the evaluation.

---

> > > > > > ### Author Response · Authors · 2023-08-20
> > > > > >
> > > > > > | Algorithm          | Energy per site | Number of layers | Hidden dimension | Bond dimension | Number of parameters (TN) | Number of parameters (NN) | Number of parameters (Total) | Runtime                        |
> > > > > > |--------------------|-----------------|------------------|------------------|----------------|---------------------------|---------------------------|------------------------------|--------------------------------|
> > > > > > | PixelCNN           | -1.74098(29)    | 7                | 48               | -              | 0                         | $5.0\times 10^5$          | $5.0\times 10^5$             | 19 hrs (A100) or 43 hrs (V100) |
> > > > > > | PixelCNN           | -1.86922(16)    | 8                | 48               | -              | 0                         | $5.7\times 10^5$          | $5.7\times 10^5$             | 59 hrs (V100)                  |
> > > > > > | PixelCNN           | -1.90021(15)    | 9                | 48               | -              | 0                         | $6.3\times 10^5$          | $6.3\times 10^5$             | 21 hrs (A100)                  |
> > > > > > | PixelCNN           | -1.90440(14)    | 10               | 48               | -              | 0                         | $7.0\times 10^5$          | $7.0 \times 10^5$            | 23 hrs (A100)                  |
> > > > > > | PixelCNN           | -1.92826(11)    | 11               | 48               | -              | 0                         | $7.7 \times 10^5$         | $7.7 \times 10^5$            | 26 hrs (A100)                  |
> > > > > > | PixelCNN           | -1.92986(10)    | 12               | 48               | -              | 0                         | $8.4 \times 10^5$         | $8.4\times 10^5$             | 30 hrs (A100)                  |
> > > > > > | PixelCNN           | -1.92869(11)    | 15               | 48               | -              | 0                         | $1.0\times 10^6$          | $1.0 \times 10^6$            | 51 hrs (A100)                  |
> > > > > > | PixelCNN           | -1.91537(13)    | 20               | 48               | -              | 0                         | $1.4 \times 10^6$         | $1.4 \times 10^6$            | 66 hrs (A100)                  |
> > > > > > | PixelCNN           | -1.88536(19)    | 8                | 72               | -              | 0                         | $1.3 \times 10^6$         | $1.3 \times 10^6$            | 35 hrs (A100)                  |
> > > > > > | PixelCNN           | -1.89266(15)    | 9                | 72               | -              | 0                         | $1.4 \times 10^6$         | $1.4 \times 10^6$            | 40 hrs (A100)                  |
> > > > > > | PixelCNN           | -1.90718(15)    | 10               | 72               | -              | 0                         | $1.6 \times 10^6$         | $1.6 \times 10^6$            | 47 hrs (A100)                  |
> > > > > > | PixelCNN           | -1.92965(13)    | 11               | 72               | -              | 0                         | $1.7 \times 10^6$         | $1.7 \times 10^6$            | 52 hrs (A100)                  |
> > > > > > | PixelCNN           | -1.91186(14)    | 12               | 72               | -              | 0                         | $1.9 \times 10^6$         | $1.9 \times 10^6$            | 57 hrs (A100)                  |
> > > > > > | PixelCNN Sign Rule | -1.92915(12)    | 11               | 48               | -              | 0                         | $7.7 \times 10^5$         | $7.7 \times 10^5$            | 31 hrs (A100)                  |
> > > > > > | PixelCNN Sign Rule | -1.93112(11)    | 12               | 48               | -              | 0                         | $8.4 \times 10^5$         | $8.4 \times 10^5$            | 34 hrs (A100)                  |
> > > > > > | PixelCNN Sign Rule | -1.93233(9)     | 15               | 48               | -              | 0                         | $1.0\times 10^6$          | $1.0\times 10^6$             | 47 hrs (A100)                  |
> > > > > > | PixelCNN Sign Rule | -1.93058(10)    | 20               | 48               | -              | 0                         | $1.4 \times 10^6$         | $1.4 \times 10^6$            | 64 hrs (A100)                  |
> > > > > > | PixelCNN Sign Rule | -1.93211(10)    | 11               | 72               | -              | 0                         | $1.7 \times 10^6$         | $1.7 \times 10^6$            | 54 hrs (A100)                  |
> > > > > > | PixelCNN Sign Rule | -1.92812(12)    | 12               | 72               | -              | 0                         | $1.9 \times 10^6$         | $1.9 \times 10^6$            | 59 hrs (A100)                  |

---

> > > > > > > ### Author Response · Authors · 2023-08-20
> > > > > > >
> > > > > > > | Algorithm          | Energy per site | Number of layers | Hidden dimension | Bond dimension | Number of parameters (TN) | Number of parameters (NN) | Number of parameters (Total) | Runtime                        |
> > > > > > > |--------------------|-----------------|------------------|------------------|----------------|---------------------------|---------------------------|------------------------------|--------------------------------|
> > > > > > > | DMRG               | -1.734326       | -                | -                | 8              | $1.2\times 10^4$          | 0                         | $1.2 \times 10^4$            | < 1 hr                         |
> > > > > > > | DMRG               | -1.866159       | -                | -                | 70             | $8.8\times 10^5$          | 0                         | $8.8 \times 10^5$            | < 1 hr                         |
> > > > > > > | DMRG               | -1.938770       | -                | -                | 1024           | $1.7\times 10^8$          | 0                         | $1.7 \times 10^8$            | 242 hrs                        |
> > > > > > > | Elementwise        | -1.94001(6)     | 7                | 48               | 8              | $1.2 \times 10^6$         | $5.0\times 10^5$          | $1.7 \times 10^6$            | 46 hrs (A100) or 89 hrs (V100) |
> > > > > > > | Blockwise          | -1.93842(42)    | 7                | 48               | 70             | $8.9 \times 10^5$         | $5.0 \times 10^5$         | $1.4 \times 10^6$            | 136 hrs (V100)                 |

---

### Official Review · Reviewer_2bnN · 2023-07-23

**Soundness:** 2 fair
**Presentation:** 2 fair
**Contribution:** 2 fair
**Rating:** 3
**Confidence:** 5

**Summary:**

This paper proposes Autoregressive Neural TensorNet (ANTN): a novel blend between Matrix Product States (MPS) and AutoRegressive Neural Network. Theoretically, this paper shows that ANTN parameterizes normalized wavefunctions, allows for exact sampling, generalizes the expressivity of tensor networks and autoregressive neural networks, and inherits a variety of symmetries from autoregressive neural networks. Experimentally, this paper shows that ANTN is more efficient than MPS and better biased than ARNN in quantum state learning, and show that ANTN outperforms MPS and ARNN in approximating the ground state of the 2D J1-J2 Heisenberg model with systems sizes up to 12x12.


**Strengths:**

The proposed architecture is novel, and importantly simple and intuitive therefore conveying a potential of enjoying “the best of both worlds” – an MPS above a very expressive AR neural network that creates a strong input representation. The core observation that one can construct such an architecture while retaining the ability to perform exact efficient sampling, is strong.
The experimental results are non-trivial. The clean experiments in 6.1 tell an interesting story of efficiency with respect to MPS and bias with respect to ARNN. In section 6.2, it is an impressive achievement to consistently surpass ARNN on the one hand and MPS on the other on the challenging J1J2 task. Moreover, the positive trend with system size is promising. Overall, the experimental results in this paper bring an appetite to try this new architecture on larger systems and on more problems.



**Weaknesses:**

Overall, I find this paper to be not well structured. My main disappointment with the writing is the lack of details on the experimental part of the paper. It is by far more convincing than the theoretical part, but unfortunately does not receive enough space and details.

For example, it is unclear to me what ANTN architecture was used for the experiments. Section 4 mentions the use of both PixelCNN and Transformer as the ARNNs underlying the ANTN. However, both tables 1 and 2 do not mention which variant was used, do not show results for both variants, and mention only the PixelCNN baseline. A long look at the experimental details in the paper body and in the appendix did not really help. It makes it very hard to assess the strength of the results when it is not clear which ARNN was used. Moreover, I am missing external performance references in order to assess the strength of the results. The field of many body quantum simulations has matured to the point where I expect new architectures to compare to external SOTA results; otherwise, it is not easy to judge results that are given relatively to baselines implemented by the authors, especially in the haziness / opacity of the current detailing of the experimental setup. Comparison to existing results on J1J2 will help convince me that the reported gains are meaningful. Besides that, much more details on the experimental setup on which the results were attained are required for this potentially important experimental outcome to be verifiable.

Theoretically, Theorem 1 is important; Section 5.2 is disappointing and does not motivate using ANTN from an expressivity standpoint; and section 5.3 is not tied to the specific experiments conducted later on is used in the remainder in the paper (so though possibly interesting it does really contribute to the practical case advocating the use of ANTN).

Elaborating, Theorem 5.2 does not provide an expressivity result that allows for reasoning regarding the benefits of ANTN. Specifically, it isn’t clear **by how much** ANTN is stronger than ARNN or MPS. The simple theoretical arguments provided do not make it clear, for example, whether ANTN is stronger than adding one more layer to the ARNN. Therefore, from an expressivity standpoint, the authors did not make a convincing case for the benefits of ANTN. Theorem 5.3 simply mentions the strength of ARNN as shown in other works, but does not contribute to the case of why ANTN is better than the existing ARNN from an expressivity standpoint. An experiment of ARNN with one more layer may very well show that the gains can also be pretty easily achieved with an existing architecture, which would be disappointing. (as a remark it isn’t clear how many layers are in the experiments on ARNN, the appendix gives some details but it isn’t clear if Transformer of PixelCNN was used).




**Questions:**

Multiple references to the “Inductive bias of the physics structure of MPS” is pretty vague. When you discuss initializing the MPS with the optimized DMRG results of the same bond dimension that kind of makes sense, but then you could also initialize the ARNN with an optimized physics related representation, so I don’t see an advantage of MPS in this “optimzed initialization” regard. As for randomly initialized MPS vs ARNN, I don’t see that a-priori the former is better suited than the latter. In particular, all of the relevant physical symmetries of ATNN listed in section 5.3 seem to be inherited from ARNN and not from MPS. On the other hand, the experiments of section 6.1 mention “wavefunctions with local or quasi-local sign structures” and “shallow random circuit to mimic the short-range interaction and generate states with sign structures”. NeurIPS is ultimately a non-physics conference and such statements are hard to decipher for me as well as for broader readership -- what exactly is the task you ran? how exactly is it related to interesting physical systems? . Especially given the important role of this point in the paper’s main argument regarding the benefits of MPS, I find it to be insufficiently explained.
Can you please elaborate and be more specific on what MPS inductive biases contribute to the physics suitability of ANTN?

Further issues:
What is h_dim? I don't see it defined.
Line 63: state-of-the-arts
Line 153: missing  punctuation after “unique”
Line 311: especially pass -> especially past


**Limitations:**

Yes

---

> ### Author Rebuttal · Authors · 2023-08-10
>
> **We appreciate the reviewer's acknowledgment that our work is theoretically novel and experimentally nontrivial. The authors apologize that due to the character limit, we are unable to respond to all the questions and have to omit some contents. The authors would provide additional explanations in the follow-up discussions.**
>
> **As the reviewer points out, our proposed architecture ANTN gives rise to a strong input representation by integrating MPS with expressive ARNN and maintains efficient sampling. Meanwhile, we have demonstrated clear advantages over MPS and ARNN with various experiments.**
>
> >**Weaknesses:**
>
> >Overall, I find this paper to be not well structured. My main disappointment with the writing is the lack of details on the experimental part of the paper. It is by far more convincing than the theoretical part, but unfortunately does not receive .....
>
> **The authors apologize for the oversight and thank the reviewer for nice suggestions on improving our writing. We will add relevant information explaining the different architectures used in different experiments in the captions as suggested. In terms of the architecture choice, all the experiments in Section 6.1 uses the Transformer and all the experiments in Section 6.2 uses the PixelCNN. The reason for such a choice is that the quantum state learning setup in Section 6.1 has a 1D structure, which makes Transformer a natural choice for the underlying ARNN, whereas in Section 6.2, the $J_1$-$J_2$ model has a 2D structure, making pixelCNN a better choice. This also makes a more fair comparison with the pure pixelCNN baseline.**
>
> >Moreover, I am missing external performance references ......
>
> **Thanks for the reviewers' suggestion and agree that comparisons to external SOTA results are necessary. In fact, we compared our results with three previous SOTA results in the original paper: RBM from NetKet, recent MPBS result, and standard DMRG result with $\chi=1024$. In Table 1, we compared our results with the RMB results for $8\times8$ system using the existing NetKet package which has been one of the SOTA packages for many models in quantum many-body systems. In addition, we also compared with one of the recent SOTA results using MPBS at $J_2=0.5$ (written in paper line 314), which is worse than our results. Moreover, DMRG has been the SOTA in this field for decades, especially under open boundary conditions which takes DMRG's advantage. Here, we benchmarked against DMRG with $\chi=1024$, which contains $10^8$ parameters, 2 orders of magnitude larger than our model. In fact, the previous SOTA MPBS did not surpass DMRG with $\chi=1024$, and it can be viewed as a milestone that our results go beyond DMRG for large systems. We will adjust the writing to make these comparisons more apparent. To clarify the details of the setup, the pixelCNN used in this work follows the implementation as mentioned in the main paper, which is implemented according to the gated pixelCNN, but slightly modified for quantum wave functions. Then, the neural networks are trained using the Adam optimizer with gradients calculated from the variational Monte Carlo algorithm (Appendix A.2), and training details are listed in Appendix C. As the reviewer suggests, we will also include additional implementation details on the experimental setup in the updated version to make sure the outcome is verifiable.**
>
> >Theoretically, Theorem 1 is important......
>
> **The authors thank the reviewer for the comment and the recognition of the importance of Theorem 1. The authors apologize that Section 5.2 writing may be too brief to highlight the expressivity of ANTN over both MPS and ARNN as an important motivation. To clarify, we have now provided detailed proof which shows that ANTN cannot be efficiently represented by either TN or ARNN to highlight this important result. (See global response.)**
>
> >and section 5.3 is not tied to the specific experiments ......
>
> **Thanks for the reviewer's comment. Section 5.3 mainly serves as an important theoretical result that the symmetries that can be implemented in ARNN can also be implemented in ANTN. As implementing symmetries on ARNN is already non-trivial, with many researchers working in this direction, explicitly developing ANTN with symmetries can be an important and promising direction for future work.**
>
> >Elaborating, Theorem 5.2 does not provide ......
>
> **The authors thank the reviewer for the comment. We have provided an updated proof above, where it is shown that a general ANTN can be written as a TN or ARNN with an exponential (in system size) number of parameters The proof demonstrates that ANTN has stronger expressivity over ARNN and TN (see global response).**
>
> >Theorem 5.3 simply mentions......
>
> **Indeed, Theorem 5.3 is a theoretical result that shows ANTN has the nice feature that it can inherit the symmetries from ARNN. The motivation here is not to show ANTN is more expressive than ARNN, but how it can non-trivially incorporate various symmetries even with tensor network on top of ARNN.**
>
> >An experiment of ARNN......
>
> **The authors thank the reviewer for the suggestion of adding more layers to the ARNN. In the global response, the authors attached additional benchmarking results for ARNN with more layers, where ARNN still lacks behind ANTN, which further supports the updated proof of Theorem 5.2. Our results clearly demonstrate that ANTN is more expressive than ARNN both theoretically and experimentally (see global response).**
>
> >(as a remark ......
>
> **The authors apologize for the confusion. All the experiments in Section 6.1 use Transformer with 2 layers, and all the experiments in Section 6.2 use PixelCNN with 7 layers. The authors will update the caption to indicate the details of neural network architectures.**
>
> >**Questions:**
>
> **The authors will answer the questions in the follow-up discussions due to the character limit.**

---

> > ### Author Response · Authors · 2023-08-10
> > **Additional Responses**
> >
> > **We apologize for not being able to include all the responses above due to the character limit. Below, we answer the questions that the reviewer raises.**
> >
> > >**Questions:**
> > >Multiple references to the “Inductive bias of the physics structure of MPS” is pretty vague.
> >
> > **Thanks for the question. In general, MPS captures the low entanglement wave function, which tends to the ground state of low dimension, and meanwhile is flexible to represent the sign structures of the wave function. In this work, we took advantage of MPS's flexibility in representing sign structures and used its representation to ground state as prior for initialization. We have added more explanation and discussion in the updated paper.**
> >
> > >When you discuss initializing the MPS with the optimized DMRG results .... MPS in this “optimzed initialization” regard.
> >
> > **The reviewer raises a good point of initializing ARNN with an optimized physics-related representation. In fact, in this work, we extensively used the transfer learning technique (mentioned in the Appendix). The pixelCNN was trained first on small systems, to learn the necessary representations easily, before training on large systems. In addition, the Marshall sign rule used in Table 1 also provides the pixelCNN with some (approximate) physics-related representation. As the table shows, this indeed improves the result in certain regimes, but it still lacks behind the ANTN. We further check if MPS initialization could improve the result of ARNN. We pretrain pixelCNN on learning DMRG results and present the results in the global response. We find that DMRG pretraining negatively affects the result. This is understandable, as learning DMRG results with complex sign structure is essentially the same task as learning a quantum state from shallow random circuit, which has been shown a difficult task for ARNN in Figure 2(b) of the original paper. In addition, since the DMRG result is just an approximation of the actual ground state, learning this state can actually make a worse initialization for pixelCNN.**
> >
> > >As for randomly initialized MPS vs ARNN......
> >
> > **Even randomly initialized MPS could still improve ANTN on its underlying ARNN, because of the flexible sign structure that the TN part of ANTN permits. To the end, DMRG is just an efficient training algorithm for MPS, which makes it desirable to train the MPS with DMRG before integrating into ANTN. However, gradient-descent-based approaches should still work when training both the TN part and ARNN part together. In the global response, we support this argument with additional results of training the ANTN without initializing the MPS with DMRG. The result shows that the ANTN performs equally well even without DMRG initialization.**
> >
> > >In particular, all of the relevant physical symmetries.......
> >
> > **Thanks for the comment. ANTN inherits symmetries from ARNN, which is a nice feature, while at the same time ANTN also inherits the flexible sign structure from MPS that does not exist in ARNN.**
> >
> > >On the other hand, the experiments of section 6.1 mention .... contribute to the physics suitability of ANTN?
> >
> > **The authors thank the reviewer for asking for clarification. To make it clear, a quantum wave function can depend on the basis in which the wave function is expressed. In a physical system, a quasi-local change of basis should not change the relevant physics of the wave function (such as long-range entanglement) but could affect the sign structure. More explicitly, a quantum wave function can be written as $\psi(x)=\sqrt{p(x)}\exp{i\theta(x)}$, with $p(x)$ a probability distribution and $\theta(x)$ a phase factor, which only takes values of $0$ and $\pi$ for real-valued Hamiltonian. The change of basis affects both $p(x)$ and $\theta(x)$, making the wave function consists of structured positive and negative values (i.e. the sign structure) that can be hard for ARNN to learn via the conditional wave functions. On the other hand, the additional bond dimension of the conditional tensors of ANTN allows the wave function to be "rotated" to the correct sign.**
> >
> > **To clarify the random circuit task, a quasi-local change of basis can be described using shallow-depth quantum circuit. In addition, shallow-depth quantum circuits can generate quantum wave functions with short-range entanglement which arises from many physical systems with local interaction. Therefore, the test on random shallow-depth quantum circuits can be viewed as a test of the neural networks for generic sign structures arising from physical systems. The details of generating the quantum wave functions under the shallow random circuit is described by Algorithm 2 in Appendix A.1. We have included more details to explain this task and the MPS inductive biases contribution to ANTN in the updated version as the reviewer suggests.**
> >
> > >**Further issues:**
> >
> > **Thanks for the corrections. We have updated the manuscript to include these changes as well as the experimental details.**

---

> > > ### Comment · Reviewer_2bnN · 2023-08-12
> > > **The advantage of ANTN over ARNN should be shown more thoroughly, both (or either) theoretically and empirically**
> > >
> > > Thank you very much for addressing many of my points.
> > >
> > > Having read your revised expressivity theorem in the global comment above, I still have a problem. The newly added part states: "A general ANTN can be written as a TN or ARNN with an exponential (in system size) number of parameters". That is not what must be shown in order to establish that ANTN is exponentially more expressive than ARNN. What must be shown in order to establish that is: "A general ANTN can **not** be written as an ARNN **unless** the latter has an exponential (in system size) number of parameters". Showing an example by construction that has exponentially many parameters does not prove the above.
> > >
> > > This resonates with an overall disagreement I seem to have with the authors on the focus of this paper and how it should be presented. As I mentioned in my original review, I think that the idea of adding an MPS over an ARNN is novel, cool, and potentially powerful. I think that the focus of a paper presenting this idea should be to study the difference specifically between ARNN and ANTN. I specifically mention this focus, because there are many papers that explore the difference between MPS and ARNN, both from theoretical and practical perspectives. As the authors note, ANTN is expected to be more powerful than MPS  precisely because ANTN builds on ARNN. So the main burden on a paper proposing ANTN should be to establish the added benefit of inclu the final MPS layer over the original ARNN. The authors' positioning includes several references to this question, but I feel it does not get the center focus as it should, and instead much focus is given to other aspects (eg, theoretical and practical advantages over MPS, which are established by design since ANTN is built over ARNN). For example, the theoretical treatment of the expressive advantage of ANTN over ARNN is not thorough enough as I point out above (I even find it misleading since it includes the word "exponential" several times, and I believe that the expressive advantage is polynomial at best when extrapolating the analysis of [1] to this case).
> > >
> > > I fully realize that establishing a sound theoretical expressivity result is hard, and I'm not sure it is required in this paper. But I also feel that the experimental part is not focused on convincing the reader that ANTN is a much better solution than ARNN. I saw that you added experiments for an ARNN with one extra layer (if I correctly understand the notation). That's a good start, why not chart the number of layers of the ARNN, and measure when it surpasses an ANTN? You are proposing a new experimental idea which augments a strong base architecture (ARNN), as a reader considering both options I'd like to understand the tradeoff.
> > >
> > > I see that some of the other reviewers like the submission as is. I am not against this idea and these results being published in NeurIPS; I am not raising my score now because I disagree with the presentation route chosen by the authors. I strongly suggest that if this paper is published in NeurIPS or a future venue that the theoretical claims are made exact and that the authors put an emphasis on a systematic empirical exploration of the difference between ANTN and the underlying ARNN.
> > >
> > > [1] Levine et al, Physical review letters 122 (6), 065301, 2019

---

> > > > ### Author Response · Authors · 2023-08-17
> > > > **Thanks for the feedback!**
> > > >
> > > > >Thank you very much for addressing many of my points.
> > > >
> > > > **Thanks a lot for the valuable feedback.**
> > > >
> > > > >Having read your revised expressivity theorem in the global comment above, I still have a problem. The newly added part states: "A general ANTN can be written as a TN or ARNN with an exponential (in system size) number of parameters". That is not what must be shown in order to establish that ANTN is exponentially more expressive than ARNN. What must be shown in order to establish that is: "A general ANTN can not be written as an ARNN unless the latter has an exponential (in system size) number of parameters". Showing an example by construction that has exponentially many parameters does not prove the above.
> > > >
> > > > **Thanks for the comment. We agree that the revised theorem does not prove that ANTN is exponentially more expressive than ARNN, which is also not what we intended to claim. We take more precautions in the wording and have added a note in the new manuscript for explicit clarification. Meanwhile, we note that it is still a nice feature that ANTN can be expanded as the sum of $\chi^N$ ARNN with $\chi$ the bond dimension and $N$ the number of qubits, even though it is possible that there could exist fewer parameters approximation of ARNN for ATNT.**
> > > >
> > > > >This resonates with an overall disagreement I seem to have with the authors on the focus of this paper and how it should be presented. As I mentioned in my original review, I think that the idea of adding an MPS over an ARNN is novel, cool, and potentially powerful. I think that the focus of a paper presenting this idea should be to study the difference specifically between ARNN and ANTN. I specifically mention this focus, because there are many papers that explore the difference between MPS and ARNN, both from theoretical and practical perspectives. As the authors note, ANTN is expected to be more powerful than MPS precisely because ANTN builds on ARNN. So the main burden on a paper proposing ANTN should be to establish the added benefit of inclu the final MPS layer over the original ARNN. The authors' positioning includes several references to this question, but I feel it does not get the center focus as it should, and instead much focus is given to other aspects (eg, theoretical and practical advantages over MPS, which are established by design since ANTN is built over ARNN). For example, the theoretical treatment of the expressive advantage of ANTN over ARNN is not thorough enough as I point out above (I even find it misleading since it includes the word "exponential" several times, and I believe that the expressive advantage is polynomial at best when extrapolating the analysis of [1] to this case).
> > > >
> > > > **We thank the reviewer for acknowledging that our work is novel, cool, and potentially powerful. We agree with the reviewer that it is important to compare ANTN with ARNN. In particular, we would like to elaborate on the comparison from both the perspectives of expressivity and physics inductive bias for optimization. In terms of expressivity, we note that ANTN has the nice feature of being written as the sum of $\chi^N$ ARNN with $\chi$ the bond dimension and $N$ the number of qubits, which shows that ANTN can be an economical way to construct more expressive quantum wave function. We fully agree with the reviewer that this does not show that ANTN is exponentially more expressive than ARNN and we have added an explicit note in the updated manuscript to avoid the confusion. Besides expressivity, we would like to highlight that the physics inductive bias, especially the sign structure discussed earlier, plays an important role in the optimization of neural network wave function. This is an important feature of L2 normalized wave function (due to the freedom of unitary transformation) which does not exist in the conventional probability generative model. It is found that the sign structure is usually challenging to be learned [Tom et al.]. In addition, the sign structure is only approximately known in very few models, or completely unknown in practice. Even for the same ARNN (i.e. same expressivity) but different sign rule, the optimization could be quite different as we show in Table.1 in the original paper. Therefore, ANTN has another important advantage over ARNN as it inherits the flexible sign structure from MPS, which is helpful for achieving stable and effective optimization without manually adding the sign rule prior (also see Table 1 in the original paper and the new data provided below). Following the reviewer's suggestion, we have added more focus on the presentation of comparing ARNN and ANTN. In particular, we added more precise discussions of the expressivity of ANTN and detailed elaborations of the physics inductive bias on the sign structure advantage of ANTN.**
> > > >
> > > > **Tom Westerhout, Nikita Astrakhantsev, Konstantin S. Tikhonov, Mikhail I. Katsnelson & Andrey A. Bagrov; Nature Communications volume 11, Article number: 1593 (2020).**

---

> > > > > ### Author Response · Authors · 2023-08-17
> > > > > **Thanks for the feedback! (Cont'd)**
> > > > >
> > > > > >I fully realize that establishing a sound theoretical expressivity result is hard, and I'm not sure it is required in this paper. But I also feel that the experimental part is not focused on convincing the reader that ANTN is a much better solution than ARNN. I saw that you added experiments for an ARNN with one extra layer (if I correctly understand the notation). That's a good start, why not chart the number of layers of the ARNN, and measure when it surpasses an ANTN? You are proposing a new experimental idea which augments a strong base architecture (ARNN), as a reader considering both options I'd like to understand the tradeoff.
> > > > >
> > > > > **Thanks for the reviewer's understanding and suggestions. We have performed additional experiments comparing ANTN with ARNN to gain more understanding on their expressivity and optimization. We have tested a series of ARNN with different layers (up to 20 layers) and hidden dimensions (up to 72 hidden dimensions) such that the largest ARNN tested has more parameters than ANTN. We found that all the ARNN energies are not as good at ANTN (which is only based on 7 layers and 48 hidden dimensions), despite that ARNNs have more parameters and took longer time to train using the same GPU (see new data below). Furthermore, we observe that an increase of parameters can help ARNN to improve energy in general, but this improvement hits a diminishing return potentially due to difficulties in optimization as the number of parameters increases. In addition, we further test the (approximate) sign rule and found that it could help ARNN to obtain better energies but still worse than ANTN without sign rule (see Table 1 in the original paper and new data below). This indicates that the ANTN has the advantage of inheriting the flexible sign structure from MPS, avoiding the manual bias of adding the sign rule. The new results have provided strong support for our expectation, both from an expressivity perspective and from the fact that ANTN has a better physics inductive bias than ARNN for optimization. We have also added a plot for the new benchmark in the updated manuscript.**
> > > > >
> > > > > >I see that some of the other reviewers like the submission as is. I am not against this idea and these results being published in NeurIPS; I am not raising my score now because I disagree with the presentation route chosen by the authors. I strongly suggest that if this paper is published in NeurIPS or a future venue that the theoretical claims are made exact and that the authors put an emphasis on a systematic empirical exploration of the difference between ANTN and the underlying ARNN.
> > > > >
> > > > > **Thanks a lot for appreciating our idea and considering that our results can potentially be published. We also thank the reviewer's suggestions on experiments which further improve the quality of our work. As we have made the theoretical claims more exact and implemented more systematic experiments comparing ARNN with ANTN, we hope that the manuscript after updates meet the reviewer's criteria and the reviewer could consider raising the score. We would also be glad to further discuss and improve if the reviewer has any additional questions or suggestions.**

---

> > > > > > ### Author Response · Authors · 2023-08-17
> > > > > > **Additional Comparisons between PixelCNN with and without Sign Rule with ANTN and DMRG at J2=0.5**
> > > > > >
> > > > > > | Algorithm          | Energy per site | Number of layers | Hidden dimension | Bond dimension | Number of parameters (TN) | Number of parameters (NN) | Number of parameters (Total) | Runtime                        |
> > > > > > |--------------------|-----------------|------------------|------------------|----------------|---------------------------|---------------------------|------------------------------|--------------------------------|
> > > > > > | PixelCNN           | -1.74098(29)    | 7                | 48               | -              | 0                         | $5.0\times 10^5$          | $5.0\times 10^5$             | 19 hrs (A100) or 43 hrs (V100) |
> > > > > > | PixelCNN           | -1.86922(16)    | 8                | 48               | -              | 0                         | $5.7\times 10^5$          | $5.7\times 10^5$             | 59 hrs (V100)                  |
> > > > > > | PixelCNN           | -1.90021(15)    | 9                | 48               | -              | 0                         | $6.3\times 10^5$          | $6.3\times 10^5$             | 21 hrs (A100)                  |
> > > > > > | PixelCNN           | -1.90440(14)    | 10               | 48               | -              | 0                         | $7.0\times 10^5$          | $7.0 \times 10^5$            | 23 hrs (A100)                  |
> > > > > > | PixelCNN           | -1.92826(11)    | 11               | 48               | -              | 0                         | $7.7 \times 10^5$         | $7.7 \times 10^5$            | 26 hrs (A100)                  |
> > > > > > | PixelCNN           | -1.92986(10)    | 12               | 48               | -              | 0                         | $8.4 \times 10^5$         | $8.4\times 10^5$             | 30 hrs (A100)                  |
> > > > > > | PixelCNN           | -1.92869(11)    | 15               | 48               | -              | 0                         | $1.0\times 10^6$          | $1.0 \times 10^6$            | 51 hrs (A100)                  |
> > > > > > | PixelCNN           | -1.91537(13)    | 20               | 48               | -              | 0                         | $1.4 \times 10^6$         | $1.4 \times 10^6$            | 66 hrs (A100)                  |
> > > > > > | PixelCNN           | -1.88536(19)    | 8                | 72               | -              | 0                         | $1.3 \times 10^6$         | $1.3 \times 10^6$            | 35 hrs (A100)                  |
> > > > > > | PixelCNN           | -1.89266(15)    | 9                | 72               | -              | 0                         | $1.4 \times 10^6$         | $1.4 \times 10^6$            | 40 hrs (A100)                  |
> > > > > > | PixelCNN           | -1.90718(15)    | 10               | 72               | -              | 0                         | $1.6 \times 10^6$         | $1.6 \times 10^6$            | 47 hrs (A100)                  |
> > > > > > | PixelCNN           | -1.92965(13)    | 11               | 72               | -              | 0                         | $1.7 \times 10^6$         | $1.7 \times 10^6$            | 52 hrs (A100)                  |
> > > > > > | PixelCNN           | -1.91186(14)    | 12               | 72               | -              | 0                         | $1.9 \times 10^6$         | $1.9 \times 10^6$            | 57 hrs (A100)                  |
> > > > > > | PixelCNN Sign Rule | -1.92915(12)    | 11               | 48               | -              | 0                         | $7.7 \times 10^5$         | $7.7 \times 10^5$            | 31 hrs (A100)                  |
> > > > > > | PixelCNN Sign Rule | -1.93112(11)    | 12               | 48               | -              | 0                         | $8.4 \times 10^5$         | $8.4 \times 10^5$            | 34 hrs (A100)                  |
> > > > > > | PixelCNN Sign Rule | -1.93233(9)     | 15               | 48               | -              | 0                         | $1.0\times 10^6$          | $1.0\times 10^6$             | 47 hrs (A100)                  |
> > > > > > | PixelCNN Sign Rule | -1.93058(10)    | 20               | 48               | -              | 0                         | $1.4 \times 10^6$         | $1.4 \times 10^6$            | 64 hrs (A100)                  |
> > > > > > | PixelCNN Sign Rule | -1.93211(10)    | 11               | 72               | -              | 0                         | $1.7 \times 10^6$         | $1.7 \times 10^6$            | 54 hrs (A100)                  |
> > > > > > | PixelCNN Sign Rule | -1.92812(12)    | 12               | 72               | -              | 0                         | $1.9 \times 10^6$         | $1.9 \times 10^6$            | 59 hrs (A100)                  |

---

> > > > > > > ### Author Response · Authors · 2023-08-17
> > > > > > > **Additional Comparisons between PixelCNN with and without Sign Rule with ANTN and DMRG at J2=0.5 (Cont'd)**
> > > > > > >
> > > > > > > | Algorithm          | Energy per site | Number of layers | Hidden dimension | Bond dimension | Number of parameters (TN) | Number of parameters (NN) | Number of parameters (Total) | Runtime                        |
> > > > > > > |--------------------|-----------------|------------------|------------------|----------------|---------------------------|---------------------------|------------------------------|--------------------------------|
> > > > > > > | DMRG               | -1.734326       | -                | -                | 8              | $1.2\times 10^4$          | 0                         | $1.2 \times 10^4$            | < 1 hr                         |
> > > > > > > | DMRG               | -1.866159       | -                | -                | 70             | $8.8\times 10^5$          | 0                         | $8.8 \times 10^5$            | < 1 hr                         |
> > > > > > > | DMRG               | -1.938770       | -                | -                | 1024           | $1.7\times 10^8$          | 0                         | $1.7 \times 10^8$            | 242 hrs                        |
> > > > > > > | Elementwise        | -1.94001(6)     | 7                | 48               | 8              | $1.2 \times 10^6$         | $5.0\times 10^5$          | $1.7 \times 10^6$            | 46 hrs (A100) or 89 hrs (V100) |
> > > > > > > | Blockwise          | -1.93842(42)    | 7                | 48               | 70             | $8.9 \times 10^5$         | $5.0 \times 10^5$         | $1.4 \times 10^6$            | 136 hrs (V100)                 |

---

### Author Rebuttal · Authors · 2023-08-10

**We would like to thank all the reviewers' comments and suggestions. We apologize that due to the character limit and math rendering issue, some explanations are simplified/omitted. We will provide additional explanations in the follow-up discussion period.**

**Reviewer 2bnN acknowledges our novel blend of MPS and ARNN for ANTN takes the best of both worlds and achieves non-trivial experimental performances. Reviewer Vtcz acknowledges that our ANTN which integrates both MPS and ARNN is "undeniably compelling". Reviewer dUKK acknowledges that our proposed ANTN absorbs the advantages of two state-of-the-art methods (MPS and ARNN) with comprehensive theoretical analysis and experimental results. Reviewer wT6D acknowledges that our approach is novel and sound for an important problem with extensive experimental support.**

**The main concerns of reviewers are on the theoretical side of our theorems and the details of experiments. We have addressed all the questions with both updated theory and new experimental data. In particular, our updated theorem 5.2 clearly demonstrates that ANTN has stronger expressivity than TN and DMRG, which is also further supported by our new experiments in terms of performance and number of parameters. (See one-page pdf)**

**We have also implemented all the writing suggestions from the reviewers and provided answers to each review. Based on our update, we appreciate if the reviewers can consider raising their scores.**

**On the theory side, reviewers expressed the concern that Theorem 5.2 is not strong enough. In response, we updated Theorem 5.2 to show that our ANTN has stronger expressivity over TN and ARNN. More specifically, a general ANTN can be written as a TN or ARNN with an exponential (in system size) number of parameters. The updated Theorem (simplified due to math rendering issue) is listed below.**

Updated Theorem 5.2.

Autoregressive Neural TensorNet has generalized expressivity over tensor network and autoregressive neural network.

Theorem:
1. Both TN and ARNN are special cases as ANTN
2. A general ANTN can be written as a TN or ARNN with an exponential (in system size) number of parameters.

Detailed proof (will add in appendix)
1. As already shown in the paper

2. (a) In ANTN, the base ARNN outputs the conditional tensors $\tilde{\psi}^{\alpha_{i-1} \alpha_i} (x_i|\boldsymbol{x_{<i}})$ to form the resulting wave functions. These conditional tensors, when explicitly written out, contain $\chi^2\cdot2^i$ elements (where $\chi^2$ comes from the two bond dimensions and $2^i$ from all the qubits at or before the current qubit $i$). Since it has been shown that ARNN has volume law entanglement while TN doesn't. The conditional tensors generated from ARNN do not permit efficient tensor decomposition. Therefore, tensor network almost has to parameterize the full conditional tensor resulting in $\sum_{i=1}^N\chi^2\cdot2^i\sim\mathcal{O}(\chi^2\cdot2^N)$ parameters in total for all qubits.
(b) In ANTN, the conditional probability is generated from the marginal probability as shown in Eq. 4. The summation over $\alpha_i$'s  contains $\chi^{2i-1}$ terms when expanded, each of which can be viewed as a (quasi-)marginal probability $q^{\alpha_1,\alpha_1',\dots,\alpha_i}(\boldsymbol{x_{\le i}})$ generated by the underlying ARNN of ANTN. Using conventional ARNN architecture, a weight matrix of shape $h_{\mathrm{dim}}\times\chi^{2i-1}\times 2$ (where $h_{\mathrm{dim}}$ is the hidden dimension, and 2 is the local dimension of the current qubit) is required to fully parameterize all the (quasi-)probabilities, leading to $\sum_{i=1}^N2\cdot h_{\mathrm{dim}}\cdot \chi^{2i-1}\sim\mathcal{O}(h_{\mathrm{dim}}\cdot\chi^{2N})$ parameters in the last layer in total.
Thus, ANTN generalizes the expressivity over both TN and ARNN and cannot be efficiently represented with either TN or ARNN.

**On the experiment side, we answer the reviewer's questions by running additional experiments and providing additional details regarding the number of parameters and training time for various algorithms in the attached PDF. The detail is summarized below.**

1. **Does increasing the number of layers allow pure pixelCNN to beat ANTN?** In attached Table R.1, we include additional results using pure pixelCNN with more layers. Although the added layer improves the result to some extend, the pixelCNN fails to beat ANTN in terms of energy calculation. This is consistent with the updated Theorem 5.2 that ARNN cannot efficiently parameterize a generic ANTN.

2. **How does the inductive bias/physics prior of MPS affects the result? Does the improvement come from the DMRG initialization or the MPS structure?** In attached Table R.1, we perform 2 addition tests: a) pretrain pure pixelCNN by learning DMRG results; b) train ANTN (elementwise) without DMRG initialization. We find that DMRG pretraining negatively affects the result, while ANTN without DMRG initialization performs as good as with DMRG initialization. This is understandable, as learning DMRG results with complex sign structure is essentially the same task as learning a quantum state from shallow random circuit, which has been shown a difficult task for ARNN in Figure 2(b) of the original paper. Since the DMRG result is an approximation of the actual ground state, learning this state actually makes a worse initialization for pixelCNN. (We note that the pixelCNN was originally initialized from the result of smaller systems sizes, as we described in Appendix C under transfer learning.)

2. **How do the runtime and number of parameters for the different algorithms compare?** In Table R.1, Table R.2, and Figure R.1, we show the number of parameters and runtime of each algorithm. In summary, DMRG has the most parameters, and is the slowest algorithm, taking as long as 2 weeks. The ANTN is much more efficient compared to DMRG while obtaining better energies, and it is comparable to pure ARNN while obtaining much better energies.

---

### Decision · Program_Chairs · 2023-09-21

**Decision:**

Accept (poster)

**Comment:**

The reviewers acknowledge the innovation to combine autoregressive (AR) models and matrix product state (MPS) for second-quantized quantum state representation, and the resulting empirical advantages over both AR and MPS models. There are also concerns on the precision of the expressiveness theorem, the explanation of conveyed inductive bias or physical structure from MPS, and completeness of empirical results to support the claims. After some discussions with the reviewers, they found additional explanations and empirical results provided in rebuttal addressed some of the concerns, and the overall opinion tends to outweigh the contributions, hence I recommend an accept for this submission.

Nevertheless, it is important that the authors include the additional explanations and empirical results in the final version of the paper. One particular issue to note is that an ARNN model could also achieve exponential expressiveness using polynomially many parameters, so the claim on the expressiveness comparison in the general rebuttal may not be fair enough.